# CDK2-mediated site-specific phosphorylation of EZH2 drives and maintains triple-negative breast cancer

Lei Nie et al.[#]

Triple-negative breast cancer (TNBC), which lacks estrogen receptor α (ERα), progesterone receptor, and human epidermal growth factor receptor 2 (HER2) expression, is closely related to basal-like breast cancer. Previously, we and others report that cyclin E/cyclin-dependent kinase 2 (CDK2) phosphorylates enhancer of zeste homolog 2 (EZH2) at T416 (pT416-EZH2). Here, we show that transgenic expression of phospho-mimicking EZH2 mutant EZH2$^{T416D}$ in mammary glands leads to tumors with TNBC phenotype. Coexpression of EZH2$^{T416D}$ in mammary epithelia of HER2/Neu transgenic mice reprograms HER2-driven luminal tumors into basal-like tumors. Pharmacological inhibition of CDK2 or EZH2 allows re-expression of ERα and converts TNBC to luminal ERα-positive, rendering TNBC cells targetable by tamoxifen. Furthermore, the combination of either CDK2 or EZH2 inhibitor with tamoxifen effectively suppresses tumor growth and markedly improves the survival of the mice bearing TNBC tumors, suggesting that the mechanism-based combination therapy may be an alternative approach to treat TNBC.

*email: mhung@cmu.edu.tw

Triple-negative breast cancers (TNBC) are highly heterogeneous and aggressive diseases and can be classified into six subtypes[1]. Among those, the basal-like breast cancer (BLBC) subtype represents a majority of TNBC. Gene expression profiling indicates that 70–80% of TNBC fall under BLBC, a highly aggressive subtype that initially responds to chemotherapy but eventually develops resistance, metastasis, and post-surgical re-occurrence[2]. The term TNBC or BLBC is used depending on the context. Because TNBC has poor outcome and a median overall survival of 10.2 months with current therapies[3], developing better strategies to improve the clinical outcome for TNBC patients is a critical unmet need.

EZH2 is overexpressed in many types of solid tumors, in particular, in TNBC[4–7]. Analysis of The Cancer Genome Atlas (TCGA) RNAseq data set[8] revealed EZH2 gene overexpression in 94.5% of basal/TNBC and 81.9% of Her2-enriched subtype (Table 1). Overexpression of genes encoding cyclin E, *CCNE1*, and *CCNE2*, also occurred at high percentages to up to 96.7% and 89.0%, respectively, in the basal-like subtype (Table 2). Moreover, 92.3% and 85.7% of BLBC co-overexpressed EZH2 and cyclin E1 or cyclin E2, respectively (Table 3). We and others have previously reported that EZH2 is phosphorylated by the cell cycle regulator CDK2 at T416[7,9,10]. CDK2 becomes activated upon binding to cyclin E[11]. Cyclin E and EZH2 are closely co-expressed in TNBC patients (Table 3), and high level of cyclin E in BRCA1-related BLBCs is associated with poor prognosis[11]. Immunohistochemically staining (IHC) of tumor specimens indicates that > 80% TNBC samples exhibit high pT416-EZH2 levels, which correlate with poorer survival[7]. Interestingly, targeted expression of wild-type EZH2 ($EZH2^{WT}$) alone in mammary glands leads to hyperplasia without developing tumors in animal studies[12,13], whereas transgenic mice harboring mammary gland-specific expression of constitutively active CDK2 develop mammary tumors containing basal-like components[14]. These findings raise an interesting possibility that overexpression of EZH2 is not sufficient to drive mammary tumorigenesis and that persistent activation of EZH2 by cyclin E/CDK2-dependent phosphorylation may be a prerequisite for tumor development. We hypothesized that CDK2-mediated site-specific phosphorylation of EZH2 drives tumorigenesis and contributes to TNBC phenotype, and inhibition of CDK2/EZH2 axis by specific inhibitors may reverse the phenotypes, thereby rendering TNBC targetable by current breast cancer therapies. CDK2/EZH2 axis blockade with specific inhibitors in clinical trials may provide new therapeutic approach against TNBC. At present, two potent and selective small molecule inhibitors of EZH2, GSK343, and EPZ-6438, are under evaluation in phase I and II clinical trials[15], with reports for the latter showing promising results[16]. EPZ-6438 selectively inhibits intracellular lysine 27 of histone H3 (H3K27) methylation in EZH2 wild-type and mutant lymphoma cells[15]. GSK343 has high selectivity for EZH2 over other methyltransferases[17].

**Table 1 Overexpression of *EZH2* in breast cancers**

| *EZH2* OE | Basal-like | HER2 | Luminal A | Luminal B | Normal-like |
|---|---|---|---|---|---|
| Sample size, n | 91 | 116 | 130 | 172 | 6 |
| Yes | 86 | 95 | 81 | 60 | 1 |
| No | 5 | 21 | 49 | 112 | 5 |
| % OE | 94.5 | 81.9 | 62.3 | 34.9 | 16.7 |

TCGA RNAseq data were downloaded from Broad Institute Firehose website (https://gdac.broadinstitute.org; BRCA cohort). An arbitrary cutoff threshold at two standard deviation (SD) above median normal expression of that gene (log2 RSEM of 8.259 for *EZH2*). Tumor with expression level above 2 SD were considered as having overexpression (OE)

**Table 2 Overexpression of *CCNE1* and *CCNE2* in breast cancers**

| | Basal-like | HER2 | Luminal A | Luminal B | Normal-like |
|---|---|---|---|---|---|
| ***CCNE1* OE** | | | | | |
| Sample size, n | 91 | 116 | 130 | 172 | 6 |
| Yes | 88 | 94 | 52 | 53 | 1 |
| No | 3 | 22 | 78 | 119 | 5 |
| % OE | 96.7 | 81.0 | 40.0 | 30.8 | 16.7 |
| ***CCNE2* OE** | | | | | |
| Sample size, n | 91 | 116 | 130 | 172 | 6 |
| Yes | 81 | 112 | 109 | 110 | 1 |
| No | 10 | 4 | 21 | 62 | 5 |
| % OE | 89.0 | 96.6 | 83.8 | 64.0 | 16.7 |

TCGA RNAseq Data set were downloaded from Broad Institute Firehose website (https://gdac.broadinstitute.org; BRCA cohort). An arbitrary cutoff threshold at two SD above median normal expression of that gene (log2 RSEM of 5.736 and 6.925 for *CCNE1* and *CCNE2*, respectively). Tumors with expression level above a cutoff were considered as having overexpression (OE)

The CDK2 inhibitor dinaciclib is currently in phase III trials and selectively inhibits CDKs, including CDK1, CDK2, CDK5, and CDK9, with different $IC_{50}$[18]. In leukemia and several types of solid tumors, dinaciclib exhibits antitumor activity as a single agent and has a safety profile in patients[19,20]. Therefore, it is practical to test our mechanism-based strategy with these ready-to-use inhibitors in preclinical mouse models. Here, we show that pharmacological inhibition of the CDK2/EZH2 axis by CDK2 and EZH2 inhibitors reactivates ERα expression, rendering TNBC sensitive to tamoxifen in vitro and in vivo.

## Results

**CDK2-mediated EZH2 phosphorylation drives tumorigenesis.** To examine whether expression of CDK2-activated EZH2 in mammary glands contributes to basal-like mammary tumor development, we established a transgenic mouse model expressing phospho-mimicking $EZH2^{T416D}$ mutant ($Tg$-$EZH2^{T416D}$) or wild-type EZH2 ($Tg$-$EZH2^{WT}$) driven by the mouse mammary tumor virus (MMTV) promoter. The transgenic mice were genotyped by PCR with two pairs of primers (Supplementary Fig. 1a). To evaluate whether the transgene is expressed and functional, mammary epithelial cells were isolated from F1 offspring of the transgenic mouse and a littermate. Immunoblotting showed EZH2 protein levels in the transgenic mammary epithelial cells were higher than that in littermate one; consistently, higher levels of H3K27 trimethylation (H3K27Me3), a catalytic product of EZH2, were observed, indicating the expressed $EZH2^{T416D}$ is active (Supplementary Fig. 1b). Histology and whole-mount analyses on the mammary glands showed that nulliparous female transgenic mice at young age (<12 months) displayed hyperplasia (Supplementary Fig. 1c, d), which was similarly observed in $Tg$-$EZH2^{WT}$ mice in earlier studies[12]. However, distinct from the $Tg$-$EZH2^{WT}$ female mice, which did not develop tumors, $Tg$-$EZH2^{T416D}$ transgenic mice developed mammary tumors with long latency (>1 year; Fig. 1a). The mammary tumor incidence was 46% (20/43) in females. In addition, the incidence of mammary tumor with lung metastasis was high (65%, 13/20). Using both basal-like (CK14) and luminal (CK18) markers, we examined the tumor tissues from $Tg$-$EZH2^{T416D}$ mice by IHC and observed CK14-positive and CK18-negative staining (Fig. 1b, left), similar to that in $p53^{-/+};Brca1^{-/-}$ tumors, which are representative of basal-like tumors (Fig. 1b, right). As expected, the luminal tumor tissues from $Tg$-$HER2/Neu$ ($Tg$-$Neu$ driven by MMTV promoter) mice showed typical positive staining for CK18 and negative for CK14 (Fig. 1b, center). These results suggested that phosphorylation of EZH2 at T416 by CDK2 is sufficient to drive mammary tumorigenesis into basal-like tumors with high frequency of lung metastasis and

| Table 3 Co-overexpression of EZH2 and cyclin E genes in breast cancers | | | | | |
|---|---|---|---|---|---|
| | **Basal-like** | **HER2** | **Luminal A** | **Luminal B** | **Normal-like** |
| *EZH2/CCNE1* OE | | | | | |
| Sample size, *n* | 91 | 116 | 130 | 172 | 6 |
| Yes | 84 | 82 | 43 | 33 | 1 |
| % Co-OE | 92.3 | 70.7 | 33.1 | 19.2 | 16.7 |
| *EZH2/CCNE2* OE | | | | | |
| Sample size, *n* | 91 | 116 | 130 | 172 | 6 |
| Yes | 78 | 93 | 75 | 54 | 0 |
| % Co-OE | 85.7 | 80.2 | 57.7 | 31.4 | 0 |

TCGA RNAseq Data set were downloaded from Broad Institute Firehose website (https://gdac.broadinstitute.org; BRCA cohort). Co-overexpression of *EZH2* and *CCNE1* or *CCNE2* was analyzed as described in Tables 1 and 2

indicated that the EZH2[T416D] transgenic mouse shares some phenotypes with the constitutively active CDK2 transgenic mouse[14].

Because tumor development was not observed in *Tg-EZH2[WT]* mice[12], we crossed *Tg- EZH2[T416D]* or *Tg-EZH2[WT]* mice with *Tg-Neu* mice, which can develop mammary tumors with luminal features (GATA3-positive, CK18-positive, and CK14-negative), to determine whether phosphorylation of EZH2 at T416 by CDK2 has a dominant role in basal-like lineage commitment in tumors. Strikingly, the double transgenic (*Tg-Neu;EZH2[T416D]*) mammary tumor tissues remained CK14-positive and CK18-negative (Fig. 1c, third panels), which were phenotypically similar to the basal-like mammary tumor tissues from *Tg-C3T*[21] control mice (Fig. 1c, bottom panels). Tumors from *Tg-Neu* and *Tg-Neu; EZH2[WT]* mice were luminal-like as indicated by CK18-positve and CK14-negative staining (Fig. 1c, upper and second panels). We further examined nine tumors from different *Tg-Neu; EZH2[T416D]* mice by IHC staining with results indicating that 78% of the tumors were strong CK14 positive and CK18 negative, and only 22% were CK18 positive (among them, one tumor displayed bi-lineage: CK18/CK14 double positive). Together, > 88% of the tumors from *Tg-Neu;EZH2[T416D]* mice were CK14 positive. These findings suggested that expression of EZH2[T416D] but not EZH2[WT] facilitates reprogramming of the lineage fate driven by HER2/Neu from luminal- to basal-like during mammary tumorigenesis. Moreover, compared with *Tg-Neu; EZH2[WT]* and *Tg-Neu* mice, *Tg-Neu;EZH2[T416D]* mice developed multiple mammary tumors with much shorter latency and survival (Fig. 1d, purple line) although the transgene expression levels were similar (Supplementary Fig. 1e).

GATA3, a luminal marker, plays functional roles in luminal-fate specification in the adult mammary glands[22,23]. Immunoblotting of the primary culture lysates of mammary tumors from *Tg-Neu* mouse showed high expression of GATA3 (Fig. 1e, lane 1). Notably, compared with *Tg-EZH2[T416D]* and other control tumor primary cultures, very low ERα protein was detected in the lysates of *Tg-Neu* tumor primary cultures at long-exposure time, which may be owing to non-tumor cells in the primary cultures. However, like the basal-like tumors from *Tg-C3T* and *p53[+/−]; Brca1[−/−]* mice, GATA3 and ERα was barely detectable in tumor primary cultures from *Tg-EZH2[T416D]* and *Tg-Neu;EZH2[T416D]* mice even after long exposure (Fig. 1e, lanes 2–3). E-cadherin and EpCAM, which are highly expressed in differentiated luminal but low in basal-like type of tumors[24], were also significantly reduced in the tumor tissues from *Tg-EZH2[T416D]* mice compared with those from *Tg-Neu* mice (Fig. 1f). In contrast, the basal-like marker SMA-α in the tumors from *Tg-EZH2[T416D]* mice was markedly increased compared with those from *Tg-Neu* mice (Fig. 1f). These results revealed that luminal markers in the *Tg-EZH2[T416D]*-derived mammary tumor cells are significantly downregulated and basal-like markers concomitantly increased,

suggesting EZH2[T416D] has a predominant role in driving basal-like mammary tumors.

The above findings raised the interesting question of whether depletion or reduction of EZH2 reverses the basal-like phenotype of the mammary tumors. To address this, we crossed the *EZH2[f/f]* conditional knockout mouse with *Trp53[f/+];Brca1[f/f];LGB-Cre* mouse[25] to generate the *p53[+/−];Brca1[−/−];EZH2[+/−]* mouse line. Compared with the *p53[+/−];Brca1[−/−]* mice whose tumor displayed basal-like characteristics, tumors from the *p53[+/−]; Brca1[−/−];EZH2[+/−]* mice exhibited CK14-positive and CK18-positive bi-lineage characteristics (Fig. 1g, right panels). Moreover, the ERα protein levels in the tumor cells from *p53[+/−]; Brca1[−/−];EZH2[+/−]* mice were significantly increased with a decrease of EZH2 protein (Fig. 1h, lane 2) compared with those from *p53[+/−];Brca1[−/−]* mice. These results indicated that deletion of just one EZH2 allele, which partially reduced EZH2 expression level in the basal-like mammary tumors, can induce ERα re-expression and switch the lineage commitment of the tumor cells.

**Activated EZH2 converts the luminal breast cancer to TNBC.** To determine whether expression of activated EZH2[T416D] is capable of reversing the luminal phenotype of breast cancer cells in vitro and in vivo, we stably expressed EZH2[WT], EZH2[T416D], or phosphorylation-deficient EZH2[T416A] mutant in the committed luminal breast cancer cell line T47D. Ectopic expression of the EZH2[T416D] but not EZH2[WT] or EZH2[T416A] significantly increased H3K27Me3 and reduced ERα protein levels (Supplementary Fig. 2a). T47D is a luminal breast cancer cell line, and CDK2/cyclin E is lower in non-TNBCs than in TNBC tumors[7] (Table 3). These results suggested that in luminal breast cancer cells, CDK2 activity is lower, and therefore, phosphorylation level of EZH2[WT] at T416 is also low, leading to less abundance of H3K27Me3. Under such condition, mutation of Thr 416 to Ala may not significantly change the global H3K27me3 level. In contrast, the EZH2[T416D] mutation mimicked T416 phosphorylation, and exhibited the highest EZH2 activity and H3K27me3 level. Importantly, T47D xenografts expressing EZH2[T416D] but not EZH2[WT] or EZH2[T416A] promoted tumor formation independently of exogenous estrogen administration (Supplementary Fig. 2b). The observed phenotype is consistent with their EZH2 methyltransferase activity, suggesting only EZH2[T416D] mutant but not WT or T416A mutant drives tumorigenesis. Next, we asked whether the tumor tissues derived from T47D-EZH2[T416D] stable cell line keep hold of the basal-like phenotype. Compared with the T47D-vector control cells, EZH2[T416D]-expressing tumor from the xenograft mouse model, along with EZH2[T416D]-expressing stable cell line and EZH2[T416D] tumor-derived cell lines (p6 and p8) displayed basal-like phenotype as indicated by the decrease in ERα, CK18, and GATA3 and increase in CK14 (Supplementary Fig. 2c). These data suggested that persistent activation of EZH2 is capable of maintaining the TNBC

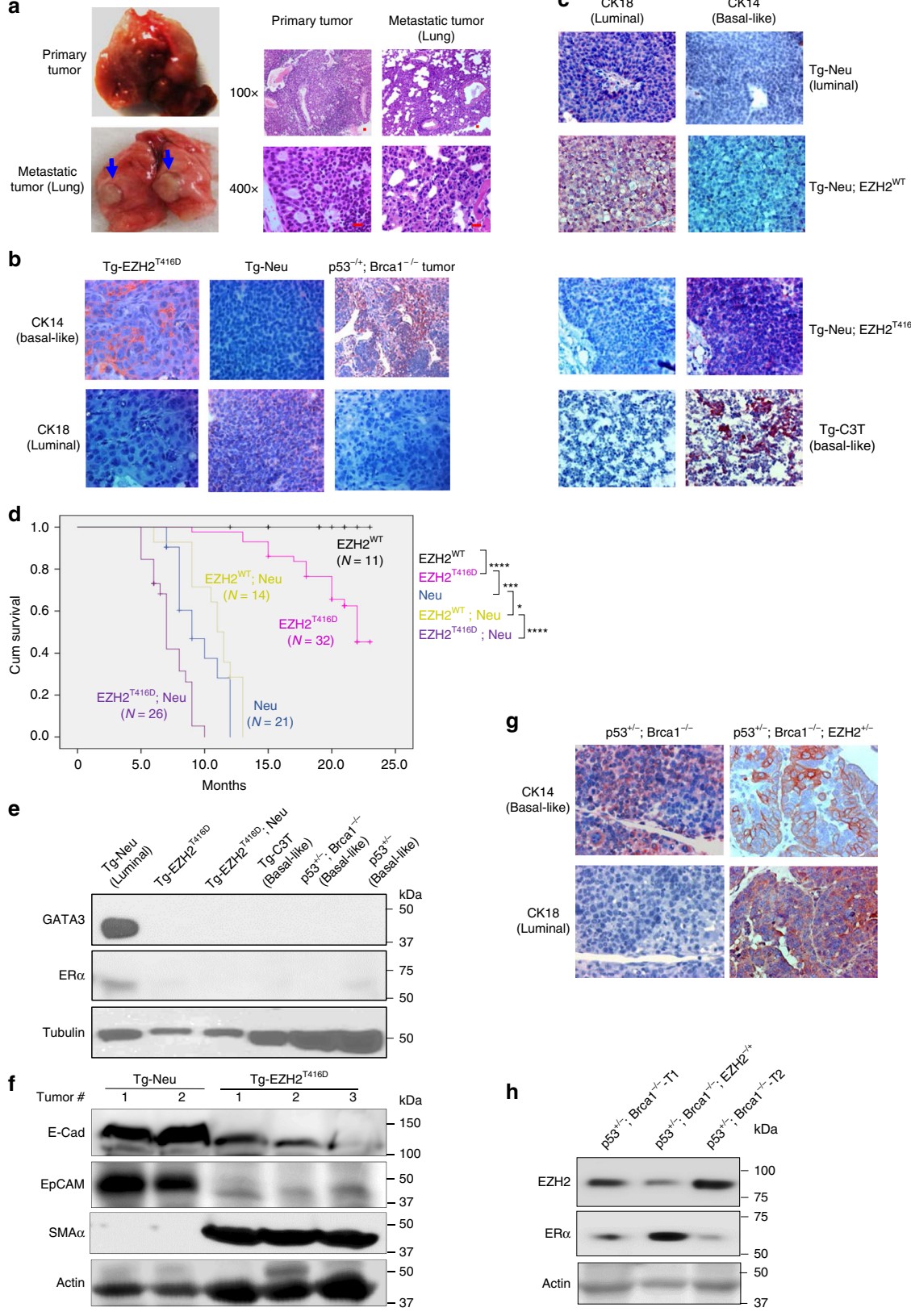

phenotype in vitro and in vivo. Next, we used a Matrigel-based 3D culture system, which can provide better microenvironment for functional assays, to further examine whether tumor cells expressing EZH2$^{T416D}$ undergo the basal-to-luminal phenotypic changes when their EZH2 activities are blocked. Indeed, treatment of the tumor cells with EZH2 specific inhibitor EPZ-6438 more significantly increased the levels of luminal markers ERα, GATA3, and CK18 in 3D cultures than in 2D culture

**Fig. 1** Expression of EZH2$^{T416D}$ mutant in mammary gland develops basal-like tumors. **a** Representative photos of the primary and lung metastatic tumors (left panel) and images of H&E staining of the tumor tissue. **b**, **c** Representative images of IHC staining of the tumors from the primary tumors of different genotypes of GEMMs. Antibodies against CK18 and CK14, the luminal and basal-like lineage marker, were used for staining, the tumor slides from Tg-Neu, p53$^{f/−}$;Braca1$^{f/f}$;LGB-Cre, and Tg-C3 mice were used for luminal and basal-like positive control, respectively. **d** Kaplan–Meier overall survival curves of Tg-EZH2$^{WT}$, Tg-Neu, Tg-EZH2$^{T416D}$, and double transgenic Tg-EZH2$^{WT}$;Neu and Tg-EZH2$^{T416D}$;Neu mice. *not significant; ***$p < 0.001$; ****$p < 0.0001$, Long-rank test. **e**, **f** Expression of EZH2$^{T416D}$ mutant in mammary epithelial cells downregulates luminal but upregulates basal-like marker proteins in the primary tumors. GATA3 and ERα in the primary tumor cell cultures **e**, and E-cadherin (E-Cad), EpCAM, and SMAα **f** in the whole-cell lysates from different genotypes of the GEMMs were immunoblotted with indicated specific antibodies; **g**, **h** conditional deletion of one-allele EZH2 in mammary gland partially reverses basal-like phenotypes of the tumor tissues. **g** Representative images of lineage marker IHC staining of the primary tumor tissue sections from the mammary tumor tissues of p53$^{+/f}$;Brca1$^{f/f}$; LGB-Cre without or with one-allele depletion of EZH2; **h** whole-cell lysates from primary tumors with genotypes of p53$^{+/−}$;Brca1$^{−/−}$; and p53$^{+/−}$;EZH2$^{+/−}$ were immunoblotted with specific antibodies as indicated. Scale bar, 20 μm. Source data are provided as a Source Data file.

(Supplementary Fig. 2d). These findings demonstrated that persistent activation of EZH2 is required for promoting tumorigenesis and maintenance of basal-like phenotype of the cancer cells and that expression of constitutively active EZH2 (EZH2$^{T416D}$) not only promotes tumorigenesis but also reprograms the committed luminal breast cancer cells into the basal-like subtype.

The above results prompted us to ask whether inhibition of EZH2 or CDK2 could reverse the TNBC phenotype, de-repress ERα gene expression, and thus render cancer cells sensitive to hormonal therapy, such as tamoxifen. To test this potentially important clinical question, we first examined the effects of EZH2 inhibitor on the induction of ERα in TNBC mouse models. In a syngeneic basal-like 4T1 mouse model, GSK343 treatment significantly inhibited 4T1 tumor growth (Fig. 2a). Interestingly, consistent with results from in vitro experiment (Supplementary Fig. 2d), GATA3 expression increased along with re-expression of ERα in 4T1 tumor tissues after GSK343 treatment for 9 days (Fig. 2b). Next, we asked whether this phenotype could be reproduced in human TNBC cell lines. MDA-MB-231 TNBC cells treated with EZH2i (EPZ-6438) induced ERα re-expression in a dose-dependent manner (Fig. 2c). Similar results were also observed in different TNBC cell lines treated with GSK343 (Supplementary Fig. 3a, b). Together, inhibition of EZH2 activities significantly reactivates ERα expression in different TNBC cell lines. As our observations in vitro and in vivo clearly demonstrated that CDK2-mediated phosphorylation of EZH2 at T416 is crucial for driving and maintaining the TNBC phenotype (Fig. 1) and that constitutively active CDK2 transgenic mice develop mammary tumors with basal-like phenotype[14], we further asked whether blocking the upstream activation of EZH2 by CDK2 inhibitor (CDK2i) can also upregulate ERα. MDA-MB-231 cells treated with a CDK2i dinaciclib re-expressed ERα and concomitantly inhibited EZH2 phosphorylation at T416 (pT416-EZH2) and H3K27Me3 in a dose-dependent manner (Fig. 2d). TNBC cells treated with CDK2i exhibited reduced phospho-CDK2 (T160) indicative of CDK2 inactivation and concomitantly increased cleaved PARP (cPARP), an evidential support of CDK2i-induced apoptosis (Fig. 2e). TNBC cell lines BT549, Hs578T (Supplementary Fig. 3a, b), and SUM-149 breast cancer cells treated with dinaciclib also exhibited upregulation of ERα expression and reduction of phospho-CDK2 and pT416-EZH2 (Fig. 2f and Supplementary Fig. 3c). In addition, reactivation of ERα expression in TNBC by CDK2i and EZH2i occurred in a time-dependent manner, and ERα expression reached a substantial level 72 hours post treatment (Supplementary Fig. 3d). Next, we further tested different EZH2i (EPZ-6438 and GSK343) and CDK2i (dinaciclib and SNS032) currently used in clinical trials, and showed that all of the inhibitors were able to re-activate ERα expression in different TNBC cell lines (Fig. 2g).

It should be noted that immunoblotting indicated low basal levels of ERα protein in the TNBC cell lines as well as a small fraction of Tg-Neu primary tumors when we used the rabbit monoclonal antibody from Abcam (ab108398) (Fig. 2c–g and Supplementary Fig. 3e). We evaluated the ERα antibody specificity by using sgRNA-mediated ERα-knockout approach and treating ERα-positive T47D cells with ER degrader fulvestrant (Ful). Both knockout and drug-induced depletion of ERα abolished the specific protein band (Supplementary Fig. 3f, g), supporting the specificity of the ERα antibody. Thus, a small quantity of ERα protein may be owing to TNBC heterogeneity[26], and the low levels of ERα protein in few Tg-Neu tumors, which are well defined as ERα-negative, may have originated from the non-malignant or premalignant cells in the tumor mass[27,28] (Supplementary Fig. 3e).

To further verify that EZH2 is phosphorylated at T416 by CDK2 in other TNBC cell lines, we performed immunoprecipitation assay with specific EZH2 and pT416-EZH2 antibodies. All TNBC cell lines examined were positive for pT416-EZH2, whose levels were reduced by dinaciclib treatment, indicating that CDK2 is responsible for the site-specific phosphorylation of EZH2 and that CDK2/EZH2-signaling axis is activated in those cells (Supplementary Fig. 3c). To further validate that EZH2$^{T416}$ phosphorylation is responsible for inhibiting ERα expression in TNBCs, we stably expressed phospho-mimic EZH2 (myc-EZH2$^{T416D}$) in BT 549 cells by lentiviral infection. The myc-EZH2$^{T416D}$ stable cell line was treated with CDK2i as described above. Enforced expression of EZH2$^{T416D}$ in BT 549 cells abolished CDK2i-induced ERα expression (Fig. 2h). These results suggested that site-specific phosphorylation of EZH2 by CDK2 contributes to the repression of ERα expression in TNBCs.

**ESR1 gene encoding ERα is a direct target of CDK2/EZH2 axis**. Next, we investigated the molecular mechanism underlying EZH2i- and CDK2i-induced ERα re-expression in TNBC. As EZH2 functions as a transcription repressor via H3K27Me3 of targeted gene promoter regions, we asked whether CDK2i and EZH2i diminish EZH2 binding to the ERα promoter regions and de-repress ERα mRNA expression. Indeed, both CDK2i and EZH2i enhanced ERα mRNA re-expression in MDA-MB-231 cells (Fig. 3a). In addition, TNBC cells treated with CDK2i or EZH2i exhibited upregulated mRNA expression of the ERα target genes, e.g., PgR, TFF1, and c-Myc (Fig. 3b–d), suggesting that CDK2i- and EZH2i-induced ERα expression in the treated cells is functional and thus targetable. We then examined the local chromatin enrichment of EZH2 on the ERα promoter and the effects of CDK2i and EZH2i on the epigenetic regulation by chromatin immunoprecipitation (ChIP) assay. The results showed a decrease in EZH2 recruitment to the ERα promoter in the presence of CDK2i or EZH2i (Fig. 3e). We also observed a

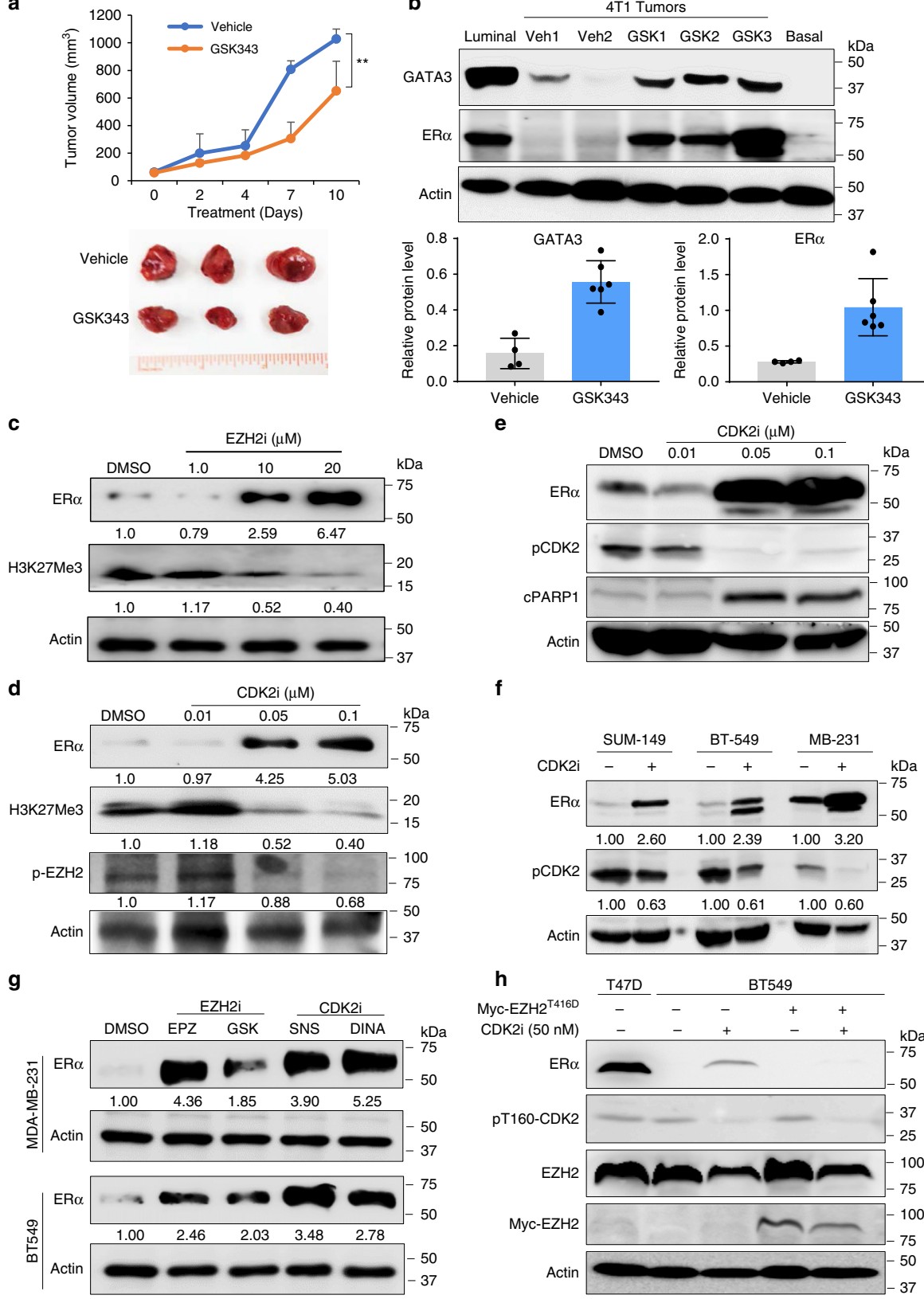

concomitant decrease in H3K27Me3 on the ERα promoter after treatment with CDK2i or EZH2i (Fig. 3f). These results indicated that inhibition of CDK2 or EZH2 blocks EZH2 binding to the ERα promoter, decreases H3K27 trimethylation, and reactivates ERα expression, leading to upregulation of its target gene expression.

**CDK2/EZH2 axis blockade sensitizes TNBC to hormonal therapy.** Hormonal therapy, such as tamoxifen, has been widely used to treat ERα-positive breast cancer[29]. However, TNBC does not respond to hormonal therapy owing to the absence of ERα expression. As the above results indicated that genetic and pharmacological inhibition of CDK2-EZH2 axis de-represses ERα

**Fig. 2** Pharmacological blockade of CDK2-EZH2 axis switches tumor cell lineage. **a** Murine basal-like cell line 4T1 cells were orthotopically transplanted on the Balb/c female mice, then treated with either vehicle or GSK343 at 100 mg kg$^{-1}$ orally daily for 9 days. Tumor volume was measured. Representative tumor samples were displayed. Points represent mean ± SEM. Data was analyzed by two-tailed Student's $t$ test: **$p < 0.05$. **b** Whole tissue lysates from different tumors treated with either vehicle (Veh1 and Veh2) or GSK343 (GSK1, GSK2, and GSK3) were subjected to western blot analysis. Cell lysates from luminal T47D cells and T47D-expressing EZH2$^{T416D}$ cells (basal-T416D) were added as positive and negative control, respectively, for GATA3 and ERα. Quantitation analysis of the images of experiments performed by using ImageQuant TL Toolbox v8.1 and normalized with loading control actin. The results are presented as bar graphs shown below. **c** Under 3D-Matrigel culture conditions, MDA-MB-231 cells were treated with different concentrations of EZH2 inhibitor, EPZ-6438 (EPZi) as indicated. Whole-cell lysates were subjected to western blot to detect ERα and H3K27me3. **d, e** MDA-MB-231 cells in Matrigel 3D culture were treated with dinaciclib (DINA) at different concentrations for 3 days. Whole-cell lysates of the treated cells were immunoblotted with specific antibodies as indicated. pT416-EZH2 (p-EZH2) was probed using specific monoclonal antibody **d**. Activated CDK2 was detected with antibody against pCDK2 (T160) **e**. **f** Different TNBC cell lines in 3D culture were treated with DINA at 50 nM for 48 hours and whole-cell lysates were used for immunoblot with indicated antibodies. **g** MDA-MB-231 and BT 549 in 3D-Matrigel culture were treated with different CDK2i, SNS032 (SNS) and DINA at 100 nM, and EZH2i, EPZ-6438 (EPZ), and GSK343 (GSK) at 25 μM and 5 μM, respectively. Whole-cell lysates of TNBCs were immunoblotted with specific antibodies as indicated. All numbers under lanes indicated the relative amount of the images normalized with actin. **h** Ectopic expressing CDK2-phospho-mimetic EZH2 mutant abolishes CDK2i-induced ERα expression. The myc-EZH2$^{T416D}$-expressing and vector control BT549 cells were treated with 50 nM DINA for 72 hours. Whole-cell lysates were immunoblotted with specific antibodies as indicated. T47D cell lysate was used for ERα control. Source data are provided as a Source Data file.

expression in TNBC, pretreating tumors with CDK2i or EZH2i may sensitize TNBC to hormonal therapy. To explore this possibility, MDA-MB-231 cells were first treated with EZH2i or CDK2i for different time periods. EZH2i and CDK2i-induced ERα re-expression reached a marked level 72 h post treatment (Supplementary Fig. 3d). Accordingly, MDA-MB-231 cells were pretreated with CDK2i or EZH2i for 72 hours followed by tamoxifen. Pre-treatment of CDK2i or EZH2i indeed rendered TNBC cells sensitive to tamoxifen and significantly reduced the sphere number and size (Fig. 4a). CDK2i or EZH2i combined with tamoxifen induced apoptosis as indicated by increased cleaved PARP proteins (cPARP; Supplementary Fig. 4a). Importantly, in vitro assays revealed synergistic effects of the combination treatment of EZH2i (CI = 0.38) or CDK2i (CI = 0.33) with tamoxifen (Fig. 4b). To further evaluate whether CDK2i/EZH2i-induced ERα protein is functional and renders TNBC cells ERα-dependent, we also tested combination of CDK2i and EZH2i with ER antagonist, fulvestrant, which binds ERα and accelerates ERα degradation. Indeed, EZH2i or CDK2i combined with fulvestrant induced strong synergistic anti-proliferative effects on TNBC cells (Fig. 4c. CI$_{EZH2i+Ful}$ = 0.20; CI$_{CDK2i+Ful}$ = 0.24.) and significantly reduced the number of viable cells compared with each inhibitor alone (Supplementary Fig. 4b). As expected, fulvestrant alone did not affect TNBC cell proliferation (Supplementary Fig. 4b). Consistently, pre-treatment of MDA-MB-231 with EZH2i or CDK2i alone induced ERα expression, but the addition of fulvestrant decreased ERα protein by promoting its degradation (Supplementary Fig. 4c). These results indicated that pharmacological inhibition of the CDK2-EZH2 axis reactivates ERα expression and that inhibitor-induced ERα protein is a functional target, rendering TNBC cells sensitive to both tamoxifen and fulvestrant.

To determine whether the combination therapy of CDK2i or EZH2i with tamoxifen is effective in vivo, we established both syngeneic and nude mouse models. In a pilot experiment, we first determined the effective dose for inducing ERα expression. Murine basal-like 4T1 cells were orthotopically inoculated into the Balb/c female mice, and were treated with EZP6438 or dinaciclib at different doses for 3 days. The pretreated tumors were then subjected to immunoblotting, which showed re-expression of ERα even at low dose (EPZ-6438, 17 mg kg$^{-1}$ and dinaciclib, 2 mg kg$^{-1}$, respectively; Supplementary Fig. 3h). Next, we tested CDK2i or EZH2i plus tamoxifen in the syngeneic mouse model. The 4T1 tumor-bearing mice were treated with either EZP6438 (34 mg kg$^{-1}$; equivalent to 102.3 mg m$^{-2}$ in human) or dinaciclib (8 mg kg$^{-1}$; equivalent to 24 mg m$^{-2}$ in

**Table 4 Inhibition of CDK2/EZH2-signaling axis reduces tumor lung metastasis**

| Group | # lung metastasis | # total mice | Metastatic rate (%) | *p* value* |
|---|---|---|---|---|
| Vehicle | 8 | 8 | 100 | – |
| TAM | 8 | 8 | 100 | NS |
| EPZ | 4 | 8 | 50.0 | 0.0253 |
| EPZ + TAM | 3 | 8 | 37.5 | 0.0090 |
| DINA | 4 | 7 | 57.1 | 0.0454 |
| DINA + TAM | 4 | 8 | 50 | 0.0253 |

*Statistical analysis was performed by using MedCalc software ("N-1" Chi-squared test)[55,56]. A *p* value < 0.05 is defined as statistically significant
*EPZ* EPZ-6438, *DINA* dinaciclib, *TAM* tamoxifen, *NS* not significant

human). The doses of EZH2i and CDK2i chosen were much lower than those of recommended phase 2 dose (RP2D: EPZ-6438, 800 mg m$^{-2}$; dinaciclib, 50 mg m$^{-2}$)[30,31]. For combination therapy, mice were administered tamoxifen after 3 days of CDKi or EZH2i monotherapy. EZH2i or CDK2i (Fig. 4d, e) monotherapy resulted in marked inhibitory effects on tumor growth compared with tamoxifen alone or vehicle control. The combination of EZH2i or CDK2i with tamoxifen induced significant tumor growth inhibition (Fig. 4d, e). Similarly, in xenograft SUM-149 mouse model, EZH2i and CDK2i monotherapy manifested significant inhibitory effects on tumor growth, and the addition of tamoxifen induced increased the suppressive effects (Fig. 4f, g). Moreover, CDK2i monotherapy resulted in a moderate beneficial effect on the survival rate of TNBC tumor-bearing mice (Fig. 4h, red dashed line, $p < 0.05$), and a significant improvement in survival when combined with tamoxifen (Fig. 4h, red solid line, $p < 0.001$). To monitor the toxic effects of the mono and combination therapies on the tumor-bearing mice (Fig. 4d, e), we examined the functional biochemical index for major organs and body weight as well after 15-day treatment schedule. The results indicated that serum levels of AST, ALT, BUN, and creatine were all within normal ranges (Supplementary Fig. 5a) with no significant body weight changes (Supplementary Fig. 5b).

Interestingly, blockade of CDK2/EZH2 axis by CDK2i or EZH2i significantly reduced distant lung metastasis in a 4T1 syngeneic mouse model (Table 4). This observation is consistent with that in recent studies in melanoma mouse models[32,33]. Compared with monotherapy of CDK2i or EZH2i,

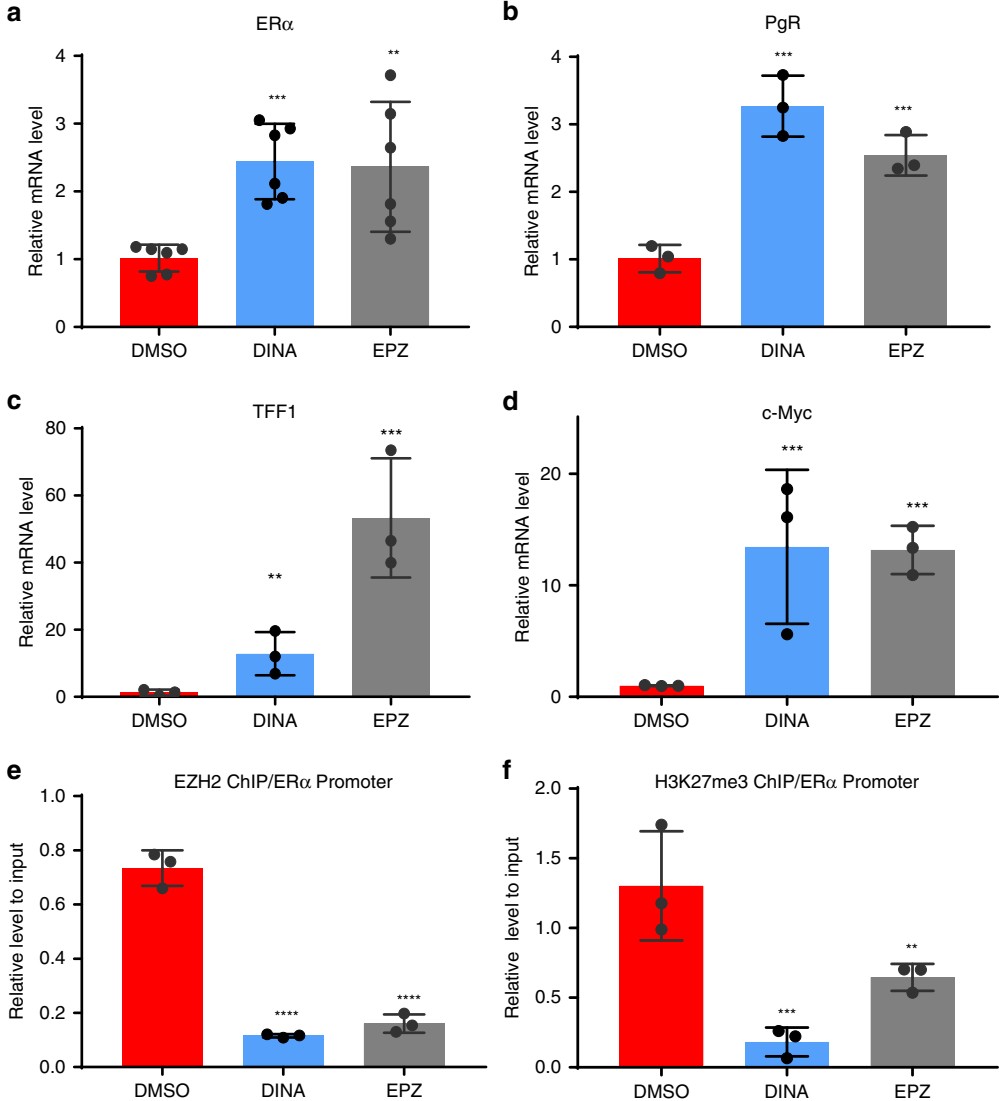

**Fig. 3** EZH2 directly targets the promoter of ERα gene (*ESR1*). **a** Under 3D culture conditions, MDA-MB-231 cells were treated with DMSO, dinaciclib (Dina) or EPZ-6438 (EPZ) for 3 days. The cells were collected to extract RNA. ERα expression was determined by real-time RT–PCR. Data were expressed as mean ± SEM ($n = 6$). **b**–**d** ERα target gene expressions were determined by real-time RT–PCR for RNA from Fig. 3**a**. Data represent mean ± SEM ($n = 3$). **e** Chromatin immunoprecipitation (CHIP) assay using anti-EZH2 antibody to detect binding of EZH2 to ERα promoter regions. The immunoprecipitates were isolated from 3D cultured cells treated with DMSO, Dinaciclib or EPZ-6438. Data are representative binding to ER promoter relative to input control (mean ± SEM, $n = 3$). **f** CHIP assay using anti-H3K27me3 antibody to detect binding of H3K27me3 to ER promoter regions. The immunoprecipitates were isolated from 3D cultured cells treated with DMSO, Dinaciclib or EPZ-6438. Data are binding to ER promoter relative to input control (mean ± SEM, $n = 3$). **p < 0.05; ***p < 0.01; ****p < 0.001, Student's *t* test. Source data are provided as a Source Data file.

the addition of tamoxifen moderately augmented the inhibitory effects on lung metastasis of TNBCs (Table 4). The reduction in distant metastasis by blockade of CDK2-EZH2 axis may partially contribute to the better survival of the tumor-bearing mice (Fig. 4h). Nevertheless, whether the combination of EZH2i or CDK2i with tamoxifen synergistically augments the anti-metastatic effects on TNBC remains to be determined by a more systematic approach with larger number of mice.

Dinaciclib has demonstrated antitumor activity in leukemia and other types of solid tumors as a single agent with a safety profile in patients[19,20]. Phase I clinical trial reports indicated that patients treated with dinaciclib at high dose (>29.6 mg m$^{-2}$, equivalent to >9.84 mg kg$^{-1}$ in mice) experienced dose-dependent adverse effects[30,31]. To rule out the possible toxicity of dinaciclib (8 mg kg$^{-1}$) combined with tamoxifen (4 mg kg$^{-1}$)

in mice for longer treatment schedule (21 day in Fig. 4f, g), we also measured the values of the indicators of liver, kidney, and heart functions, and the mouse body weight for the combined regimen. No significant toxicities were observed in animals treated with the combination of 8 mg kg$^{-1}$ CDK2i and 4 mg kg$^{-1}$ tamoxifen for 21 days (Supplementary Fig. 5c, d).

To minimize potential chronic adverse effects, we next examined the therapeutic efficacy of low dose dinaciclib for the combination treatment in one of the most malignant TNBC xenograft mouse models. Mild inhibition of tumor growth was observed in MDA-MB-231 tumor-bearing mice treated with dinaciclib (2 mg kg$^{-1}$, equivalent to 6 mg m$^{-2}$ in humans; Supplementary Fig. 6a, blue line). In contrast, adding tamoxifen augmented the inhibitory effects of dinaciclib on tumor growth ($p < 0.01$, Supplementary Fig. 6a, red line). As mentioned above,

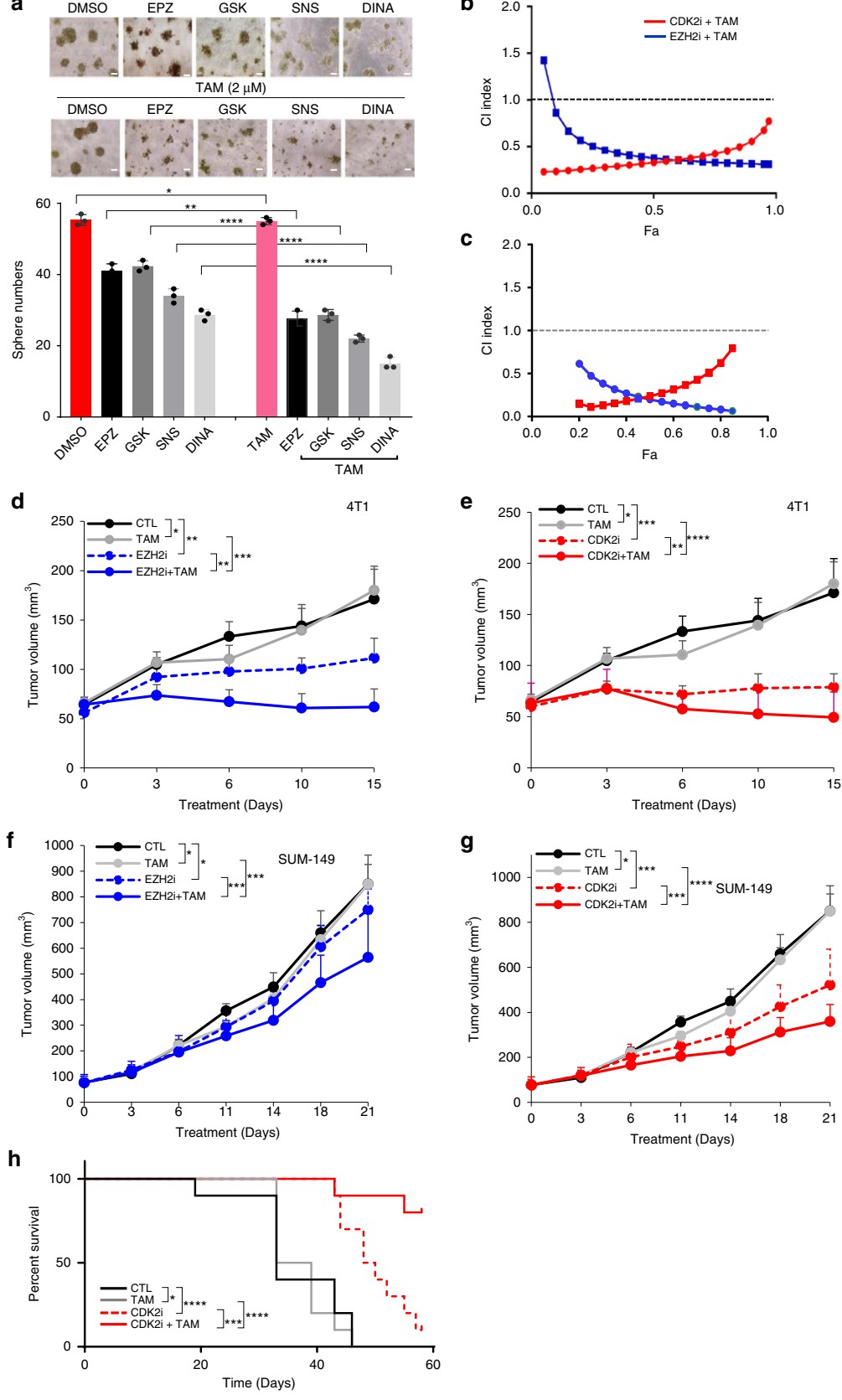

**Fig. 4** CDK2/EZH2 blockade creates combination regimen with tamoxifen against TNBC. **a** Effective inhibition of TBNCs in 3D cultures by combination treatment of CDK2/EZH2 inhibitors (EPZ, GSK; SNS, and DINA) and TAM determined by 3D-Matrigel sphere formation assays. Representative images (upper panel) and bar graph of sphere numbers (bottom panel). Scale bar, 200 μm. **b, c** Synergistic effect of combination treatment of the TNBCs in vitro with CDK2/EZH2 inhibitors and TAM **b** or fulvestrant, Ful **c**. **d, e** Inhibitory effect of the combination therapy of CDK2 or EZH2 inhibitor and tamoxifen (TAM) on tumor growth in syngeneic mouse model. 4T1 cells were orthotopically transplanted on Balb/C female mice. The tumor-bearing mice were first pretreated with either EZH2i (EPZ-6438; EPZ) at 34 mg kg$^{-1}$ or CDK2i (dinaciclib) at 8 mg kg$^{-1}$ daily for 3 days, followed by combined with 4 mg kg$^{-1}$ TAM for more than 2 weeks. The tumor burdens were measured with digital caliper ($n = 10$). *not significant, **$p < 0.05$, ***$p < 0.01$, ****$p < 0.001$, Student's $t$ test. **f, g** The effects of EZH2 (**f**) or CDK2 (**g**) inhibitor alone or in combination with TAM on TNBC tumor growth in a xenograft mouse model. TNBC SUM-149 cells were orthotopically inoculated on nude mice. The tumor-bearing mice were treated and monitored as described above **d, e**. EZH2i (GSK343) and CDK2i (dinaciclib) were administered at a dose of 100 mg kg$^{-1}$ and 8 mg kg$^{-1}$, respectively ($n = 10$). *not significant; ***$p < 0.01$; ****$p < 0.001$, Student's $t$ test. **h** Survival curves of tumor-bearing mice treated with CDK2i, TAM, or their combination ($n = 10$). *not significant, **$p < 0.05$, ****$p < 0.001$, Log-rank test. Source data are provided as a Source Data file.

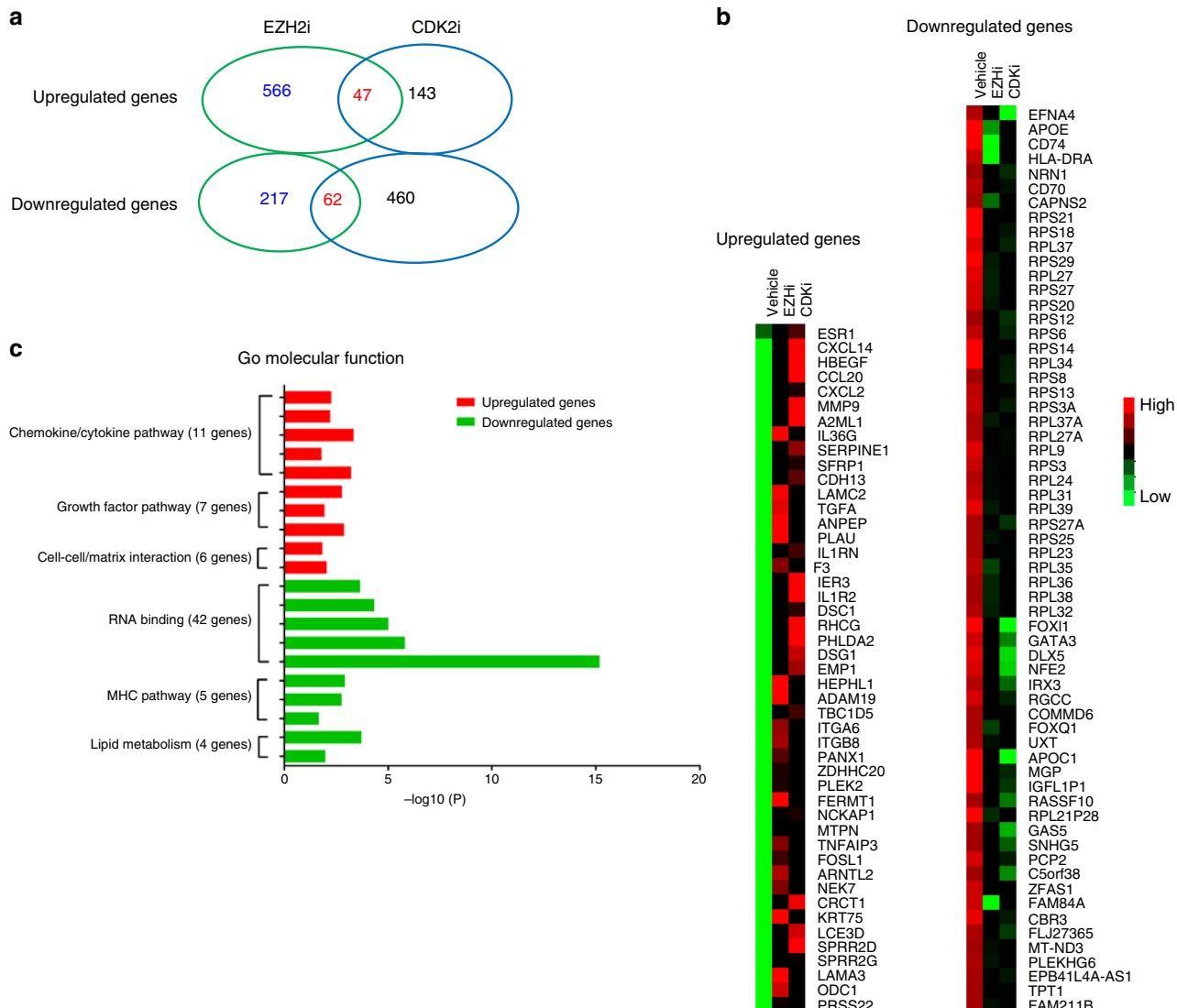

**Fig. 5** Transcriptome profiles in TNBC regulated by the CDK2/EZH2 axis. The xenograft tumors of SUM-149 cells were treated with vehicle, CDK2i and EZH2i, respectively. Total RNAs were isolated from the tumors by using RNeasy Mini Kit (Qiagen). Differential expression analyses of these tumors were performed by using *Homo sapiens* (GRCH37/hg19) as reference genome and genome annotation, and setting fold change to 2 and $p < 0.05$ as cutoff when using Limma package to call differential gene. **a** Venn diagram of the differential and commonly shared genes regulated by CDK2 and EZH2. **b** Representative heat map of top-regulated genes by blockade of CDK2/EZH2 signal axis. **c** Representative top-shared upregulation and downregulated genes and their involved pathways. Enrichment analysis was performed as described[58]. Source data are provided as a Source Data file.

the first dose-limiting toxicity (DLT) of dinaciclib occurred at 29.6 mg m$^{-2}$ (equivalent to 9.84 mg kg$^{-1}$ in mice) in a phase 1 clinical trial[30]. Our data showed that tumors from mice treated with dinaciclib at a dose (2 mg kg$^{-1}$) lower than DLT exhibited reactivation of ERα expression (Supplementary Fig. 3h, right panel). We also observed modest but statistically significant effects on tumor growth for the dinaciclib-tamoxifen combination (Supplementary Fig. 6a, red line, $p < 0.01$). Moreover, compared with vehicle control, the combination of low dose dinaciclib (2 mg kg$^{-1}$) and tamoxifen (4 mg kg$^{-1}$) increased the survival rate of 60% of tumor-bearing mice (Supplementary Fig. 6b, red line, $p < 0.05$), suggesting this minimal dose combination regimen is sufficient to repress tumor growth and significantly improve survival. Together, these results suggested that EZH2i or CDK2i combined with tamoxifen is a potentially effective therapeutic strategy against TNBC.

**CDK2/EZH2 axis orchestrates global transcriptome in TNBC.** The genetic evidence has demonstrated that CDK2 is indispensable for the proliferation and survival of most cell types[34]. Specific phosphorylation of Myc by CDK2 was recently reported to be required for promoting Myc-driven tumorigenesis and progression by suppressing senescence pathways[35,36]. Here, we first demonstrated that site-specific phosphorylation of EZH2 by CDK2 is required for tumor cell lineage commitment in tumorigenesis. To drive tumorigenesis and determine lineage type, the cells should be accompanied by epigenetic remodeling, leading to alternations of gene expression profile. Therefore, we sought to determine the changes of whole exomes treated with the specific CDK2i/EZH2i in addition to the lineage markers and ERα mentioned above. To this end, we performed RNAseq analysis to determine the global transcriptome alterations in SUM-149 xenograft tumors treated with CDK2i or EZH2i. Differential expression analysis of these tumors showed that more than 801 and 741 genes were affected by CDK2i and EZH2i treatment, respectively. Among those, we identified top 109 commonly upregulated and downregulated gene sets in the inhibitor-treated tumors (Fig. 5a, b; RNAseq data in the GEO repository, GSE132194). Enrichment analysis indicated that the upregulated genes were mostly involved in (1) chemokine/cytokine pathway, (2) growth factor pathways, and (3) cell-cell/matrix interaction, and downregulated genes in (1) RNA binding, (2) major histocompatibility complex (MHC) pathways, and (3) lipid metabolism (Fig. 5c). Importantly, among these commonly altered genes, the expression of at least three ERα-regulated genes, *SPRR2D*, *SPRR2G*, and *LCE3D*, were significantly increased (Fig. 5b; Supplementary Dataset, GSE132194). It is worth noting that the *RPL39* gene encoding ribosomal protein L39 was downregulated in CDK2i- and EZH2i-treated tumors (Fig. 5b). *RPL39* is associated with therapy resistance, metastasis, and cancer stem cell self-renewal of TNBCs[37,38]. In total, the RNAseq results provided additional evidence to support our hypothesis that epigenetic regulator EZH2 is one of the key targets of CDK2, and that activation of EZH2 by CDK2-mediated phosphorylation results in global transcriptome changes in addition to ERα and markers indicative of luminal and basal-like lineages (Supplementary Fig. 6c).

## Discussion
EZH2 mutation or overexpression frequently occurs in lymphoma, leukemia, and several solid tumor types, such as breast, prostate, and bladder cancers, and melanoma[39]. Therefore, EZH2 is considered as a tumor driver. However, transgenic expression of wild-type EZH2 in mammary and prostate glands does not lead to tumor development[12,13]. We presented genetic evidence to demonstrate that CDK2-mediated phosphorylation of EZH2 at T416 is critical for its role as a tumor driver and lineage indicator. Notably, EZH2i alone or in combination with tamoxifen was less effective than CDK2i on tumor growth suppression in the mouse models. Consistently, ERα-induction in the tumors treated with EZH2i at the current dose is not as effective as CDK2i, suggesting that targeted inhibition of pT416-EZH2-containing complex is more effective than unbiased inhibition of all types of EZH2-containing complexes to de-repress ERα promoter activity. Alternatively, the dose of EZH2i used in our experiments might have been too low. By contrast, in phase 1 clinical trials, the RP2D dose of EPZ-6438 is 800 mg m$^{-2}$, equivalent to 259.46 mg kg$^{-1}$ in mice[16]. A phase 1 study of EPZ-6438 showed that EZH2i is effective and well-tolerated in patients[16]. Therefore, further work would be required to identify an optimal dose of the EZH2i and tamoxifen combination for TNBC.

Importantly, our findings demonstrated that pharmacological inhibition of the CDK2-EZH2 axis by EZH2i and CDK2i at a lower dose than RP2D induced ERα expression in the committed TNBC cells, rendering them targetable by hormonal therapy. Interestingly, a recent study showed that EZH2 inhibitors upregulated androgen receptor expression and sensitized the prostate cancers to anti-androgen therapy[40]. In addition, EZH2 activity was implicated in chemo-resistance of small cell lung cancers, and epigenetic changes in chemo-resistance were reversible upon EZH2i treatment[41]. Together, our current study suggested that the CDK2-EZH2 axis is an actionable target in different types of solid tumors, and that CDK2i or EZH2i combined with chemotherapy or hormonal therapy may be a potential therapeutic strategy to effectively treat TNBC and other solid tumors with EZH2 overexpression and high pT416-EZH2. A phase I study demonstrated that the combination of dinaciclib and chemo drug epirubicin in patients with metastatic TNBC did not provide additional benefits at least partially owing to substantial toxicity[42]. Notably, at much lower doses of CDK2i used in the current study than in clinical trials[30,42], we did not observe any significant acute cytotoxic effects, but the survival of tumor-bearing mice was markedly improved (Fig. 4h and Supplementary Fig. 5). Tamoxifen is widely and effectively used to treat ERα-positive breast cancer with tolerable toxicity[29], and no acute toxicity for tamoxifen monotherapy or in combination with EZH2i or CDK2i was observed using the doses we have chosen.

Previous studies have shown that DNA methyltransferase inhibitor (DNMTi) and histone deacetylase inhibitor (HDACi) re-activate ERα expression in TNBC[43–45], and a phase II clinical trial of DNMTi (5-azacytidine) and HDACi (entinostat) with tamoxifen for advanced hormone-resistant and TNBC was proposed. However, preliminary results indicated that none of the 13 TNBC patients showed partial response to the treatment[46]. Further analysis revealed that DNMTi plus HDACi treatment did not induce ERα expression in TNBC patients[46]. Cyclin E/CDK2 and EZH2 are overexpressed in TNBC and pT416-EZH2 protein level is high in >80% TNBC patient samples[7,11], suggesting that inactivation of DNMT1 may be not sufficient to de-repress ERα gene expression because phosphorylated EZH2 by CDK2 potentially binds to and silences the ERα gene promoter in most of TNBC patients. HDAC inhibition does not induce ERα in human TNBC cell lines or in PDX mouse model[47]. Given that the presence of EZH2 is necessary for the binding of DNMT to EZH2-target promoters but not vice versa[48] and that persistent activation of the CDK2-EZH2 axis in most of TNBC cells could at least partially attenuate ERα re-expression induced by inhibition of DNMT or HDAC, targeting the terminal mediator EZH2 of ERα gene expression is likely a prerequisite for reactivating ERα expression in TNBC. Roswall et al.[49] recently reported that interruption of platelet-derived growth factor-CC signaling

between cancer cells and the microenvironmental stroma converts BLBC into ERα positive, leading to enhanced sensitivity to hormone therapy. Their findings provided additional evidence to support our hypothesis that converting TNBC into ERα-positive breast cancers is a therapeutically actionable approach for TNBC patients.

RNAseq analysis showed that CDK2i- and EZH2i-upregulated genes were enriched in the chemokine/cytokine pathways, such *as TNFA, CXCL2*, and *CCL20*, which affect the tumor microenvironment[50]. Notably, cytokine IL-36γ was upregulated by CDK2i/EZH2i, and tumor cell-secreted IL-36γ exhibits profound antitumor effects and can convert the tumor microenvironment into one in favor of tumor eradication[51]. *RPL39*, one of the downregulated genes by CDK2i/EZH2i, is associated with metastasis and survival of TNBC patients[37,38]. Therefore, CDK2i and EZH2i-attenuated lung metastasis (Table 4) could be partially owing to the repression of *RPL39* gene expression. Nevertheless, the biological functions of the commonly targeted genes of CDK2 and EZH2 remain to be further investigated.

## Methods

**Reagents and plasmids**. EPZ-6438 and Dinaciclib were obtained from Selleck Chemicals (Houston, TX). SNS032 was purchased from Calbiochem. GSK343 was purchased from AdooQ. pT416-EZH2 monoclonal antibody was raised by immunization of mouse with phospho-peptides (ANSRCQ-pT416-PIKMKPNIE). The site-directed mutagenesis of EZH2 to generate EZH2$^{T416A}$ and EZH2$^{T416D}$ mutants was performed according to the manufacturer's protocol (Clontech; Palo Alto, CA) as previously described[7]. sgRNA against ERα was obtained from Thermo Fisher. Detailed information of the antibodies used in the experiments can be found in Supplementary Table 1.

**Cell culture**. All cell lines were obtained from the ATCC (Manassas, VA) and independently validated by STR DNA fingerprinting at MD Anderson, and tests for mycoplasma contamination were negative. These cells were grown in DMEM/F12 or DMEM medium supplemented with 10% fetal bovine serum, penicillin, and streptomycin.

**Generation of EZH2 mutant transgenic mouse model**. *MMTV-EZH2$^{T416D}$ (Tg-EZH2$^{T416D}$)* transgenic mouse was generated by cloning a cDNA harboring T416D mutation into the *Eco*RI/*Hin*dIII of plasmid MMTV-SV40-BSSK. The cDNA of human EZH2 was sequenced prior to digestion with *Spe*I and *Sal*I. The purified DNA fragments containing the expression cassette of EZH2 mutant was injected into one-cell zygotes of FVB/N mice performed in the Genetically Engineered Mouse Core facility, MD Anderson Cancer Center. *MMTV-EZH2$^{WT}$ (Tg-EZH2$^{WT}$)* mice[12] were generated similarly as described above. The transgenic founder mice were confirmed by PCR-based genotyping and western blot analysis of mammary epithelial cells with EZH2 and H3K27Me3 antibodies. Genotyping of the EZH2$^{T416D}$ transgene was performed by PCR with two pairs of primers (5′–3′):
Pair 1 Forward: GGAACCTTACTTCTGTGGTGT
Reverse: GCA TTCCACCACTGCTCCCATTC
Pair 2 Forward: CTGATCTGAGCTCTGAGT
Reverse: GACAAACAATTCCAACAGTC
PCR reaction: 95˚C, 30 s; 58 ˚C, 30 s, 72 ˚C, 1 min, up to 30 cycles.

**Three-dimensional (3D) cell culture**. 3D culture was performed as previously described[52]. In brief, prechilled culture dishes were coated with a thin layer of cold growth factor-reduced Engelbreth-Holm-Swarm extracellular matrix extract (Matrigel, BD Biosciences, San Jose, CA). Dishes were then incubated for 30 min at 37 °C to allow EHS gelation. The cultured TNBC cells were trypsinized and resuspended in the medium containing 2.5% EHS. The mixture of cells in EHS medium were pipetted onto the gel surface, and incubated at 37 °C for 3 days. Drugs were added to the prechilled medium containing 2.5% EHS and to the cells in 3D-Matrigel culture. The cell extraction from 3D-matrigel cultures was performed by PBS-EDTA.

**Quantitative real-time RT–PCR**. Total RNA was extracted from the treated cells in 3D-Matrigel culture with Trizol (Invitrogen). Quantitative RT–PCR (qRT–PCR) was performed using SYBR Green dye on a Bio-Rad PCR machine[53]. In brief, 1 μg of total RNA was reverse transcribed into cDNA using SuperScript III (Invitrogen) in the presence of random hexamers. All PCR reactions were performed in triplicate with SYBR Green Master Mix (Applied Biosystems) in the presence of 1 μM of both the forward and reverse primer according to the manufacturer's recommended thermocycling conditions, and then subjected to melt-curve analysis. The quantity of the target gene for each sample was divided by the average sample

quantity of the GAPDH to obtain the relative level of the gene expression. The primer sequences (5′–3′) for the transcripts analyzed are as follows:
*ESR1* forward: CTCTCCCACATCAGGCACA
*ESR1* reverse: CTTTGGTCCGTCTCCTCCA
*PGR* forward: CTTAATCAACTAGGCGAGAG
*PGR* reverse: AAGCTCATCCAAGAATACTG
*TFF1* forward: GTCCCCTGGTGCTTCTATCC
*TFF1* reverse: GAGTAGTCAAAGTCAGAGCAGTCAATCT
*c-myc* forward: TTCGGGTAGTGGAAAACCAG
*c-myc* reverse: CAGCAGCTCGAATTTCTTCC
PCR reactions were carried out as described above.

**ChIP assays**. ChIP assay was carried out with the EZ-ChIP kit (Millipore) as previously described[53] with antibodies against EZH2 and trimethyl-H3K27. In brief, MDA-MB-231 cells in 3D-Matrigel culture were treated with CDK2 and EZH2 inhibitors for 72 h. Cells were then cross-linked for 10 min by adding formaldehyde to a final concentration of 1%. The cross-linking reaction was stopped by addition of 1/20 volume of 2.5 M glycine followed by cell lysis and sonication. The lysates were incubated with specific antibody overnight at 4 °C. Reversal of cross-linking was performed at 65 °C for 3 hours. The purified DNA was analyzed by qPCR. The primer sequences (5′–3′) for the ERα promoters are as follows:
ERα forward: TCTGCCCTATCTCGGTTACA
ERα reverse: TAAAGGTGGCCCAGGGAAGA
PCR reaction: 95°C, 30 s; 55 °C, 30 s, 72 °C, 1 min, up to 30 cycles.

**Immunohistochemistry and immunoblot**. Immunohistochemistry was performed as described previously[54]. In brief, tumor samples were deparaffinized and rehydrated. Antigen retrieval was carried out by heating in 0.01 M sodium citrate buffer (pH 6.0). Endogenous peroxidase activities were blocked by 1% hydrogen peroxide. To prevent nonspecific staining, 10% normal serum was used. The samples were incubated with primary antibodies at 4 °C overnight. Biotinylated secondary antibody and avidin-biotin peroxidase complex solution were incubated for 1 h at room temperature. Color was developed with the 3-amino-9-ethylcarbazole solution. Counterstaining was carried out using Mayer's hematoxylin. Whole-cell lysates of cell lines and tumor tissues were used for chemiluminescent western blot assays (ECL kit, Cat # 32209, Thermo Fisher) with ImageQuant LAS 40000 (GE Health BioScience AB). The lysate proteins were separated by SDS-polyacrylamide gel electrophoresis, followed by immunoblot with the specific antibodies. Quantitation analyses of the images were performed by using ImageQuant TL Toolbox v8.1 and normalized with loading control actin or tubulin.

**Primary mammary tumor cell cultures**. The primary tumors from different genotypes of GEMMs were mechanically sliced into small pieces smaller (< 1 mm$^3$) using scalpels and then dissociated with collagenase and hydrogenase. The resultant organoids of the tumors were collected and cultured on the collagen I (EMD Millipore)-coated six-well plates. The primary tumor cells were cultured in DMEM/F12 Ham's medium supplemented with 5% horse serum (Life Technologies), 10 μg ml$^{-1}$ insulin (Invitrogen), 20 ng ml$^{-1}$ EGF (Sigma), 100 ng ml$^{-1}$ cholera toxin (EMD Millipore), 0.5 mg ml$^{-1}$ hydrocortisone (Sigma), and antibiotic-antimycotic cocktail (Gibco).

**Transcriptome analysis by RNA sequencing**. Total RNAs were isolated from the SUM-149 transplant tumor tissues treated with vehicle, EZH2i or CDK2i by using the RNeasy Mini Kit (QIAGEN). RNAseq was performed by NOVOGENE (UC Davis) and *Homo sapiens* (GRCH37/hg19) was used as reference genome annotation. The top 109 commonly upregulated and downregulated genes are shown in Supplemental Dataset. RNAseq data is available to the public (GSE132194; NCBI tracking system #20042815).

**TCGA data set analysis**. To identify TCGA breast tumors with EZH2 and cyclin E gene overexpression, we used data from normal breast tissue in TCGA and set up a cutoff at two standard deviation (SD) above median normal expression of EZH2. RNAseq data set includes a total breast cancer samples and normal breast tissues. The breast cancer subtypes were defined by the PAM50 gene signature. TCGA RNAseq data were downloaded from Broad Institute Firehose website (https://gdac.broadinstitute.org; BRCA cohort). A total of 515 breast cancer samples with tumor subtype information and 102 normal breast tissues[8] were used in our analysis. To call gene overexpression in breast tumors, we used an arbitrary cutoff threshold at two standard deviation above median normal expression of that gene (log2 RSEM of 8.259, 5.736, and 6.925 for EZH2, CCNE1, and CCNE2, respectively). Tumor with expression level above a cutoff were considered as having an overexpression.

**Animal studies**. Tumorigenesis assays were performed in an orthotopic breast cancer mouse model. T47D cells ($1 \times 10^6$) with lentiviral-stable expression of EZH2$^{WT}$, EZH2$^{T416A}$, or EZH2$^{T416D}$ were injected into mammary fat pads of female NOD/SCID mice (six per group). For MDA-MB-231 and SUM-149 orthotopic xenograft mouse model, $1 \times 10^6$ cells were mixed with Matrigel at 1:1

ratio and injected into mammary fat pads of *Nu/Nu* female mice. For 4T1 syngeneic mouse model, $1 \times 10^5$ cells mixed with Matrigel were used for orthotopic injection. Tumor size was measured every 3 days with a digital caliper, and tumor volume was determined using the formula: $L \times W^2/2$, where $L$ is the longest diameter and $W$ is the shortest diameter. Tumor-bearing mice were pre-treated with EPZ-6438 (EPZ) orally daily or dinaciclib by i.p. injection daily for 3 days followed by combination treatment of EPZ or dinaciclib and tamoxifen. MMTV-*EZH2*$^{T416D}$ transgenic mouse was generated by cloning the EZH2 cDNA harboring T416D mutation into *Eco*RI/*Hin*dIII sites of the plasmid MMTV-SV40-BSSK. The cDNA was sequenced prior to digestion with *Spe*I and *Sal*I. The purified DNA fragments containing the expression cassette of the EZH2 mutant was microinjected into one-cell zygotes of FVB/N mice performed in the Genetically Engineered Mouse Core Facility at MD Anderson Cancer Center. MMTV-*EZH2*$^{WT}$ mice were generated as previously described[12]. The transgenic founder mice were confirmed by PCR-based genotyping and Western blot analysis. *EZH2*$^{flox/flox}$ mouse was kindly provided by Dr. Alexander Tarakhovsky at The Rockefeller University (MMRRC_015499-UNC). The *p53*$^{LoxP}$ mouse line was generously provided by Dr. Anton Berns (NKI-AVL). *Trp53*$^{tm1Brd}$;*Brca1*$^{tm1Ash}$;*Tg (LGB)-Cre)74Acl/J*, *Tg(MMTV-Cre)4Mam/J* and *FVB/N-Tg(MMTV-Neu)202Mul/J* mouse lines were obtained from the Jackson Laboratory (Bar Harbor, ME). *FVB-Tg (C3-1-Tag)cJeg* mouse and primary tumor culture was generously provided by Dr. Khandan Keyomarsi. Mouse mammary gland tissues and tumors for histological analysis were fixed overnight in 10% neutral buffered formalin (Sigma-Aldrich), processed through alcohols and xylene, and embedded in paraffin. The tissues slides were then stained with hematoxylin and eosin, and the indicated antibodies.

**Study approval**. All experimental procedures involving animals in this study were reviewed and approved by The University of Texas MD Anderson Cancer Center Institutional Animal Care and Use Committee (IACUC; Protocol Numbers 00001245-RN01 and 00001334-RN01). This study did not involve any human tissue samples.

**Statistical analysis**. Student's *t* test (two-sided) and $\chi^2$ test were used to compare difference between two groups, and a comparison of proportions between two groups was performed by using MedCalc software[55,56]. For survival analysis, log-rank (Mantel–Cox) test was performed using the GraphPad Prism 7 software. Synergistic effect and CI values in vitro were calculated using Chou–Talalay method[57]. A *p* value of <0.05 was considered statistically significant.

**Reporting summary**. Further information on research design is available in the Nature Research Reporting Summary linked to this article.

## Data availability
All data supporting the findings of this study are available with the article, and can also be obtained from the authors. RNAseq data has been deposited in the NCBI GEO repository with an accession ID GSE132194 [https://www.ncbi.nlm.nih.gov/geo/query/acc.cgi?acc=GSE132194].

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

## Acknowledgements

We thank Dr. Paul B. Koller for critically reviewing this manuscript and helpful advice, and the GEMM core facility at MD Anderson Cancer Center (CCSG CA016672) for assistance in generating and breeding the mice. We are grateful to Dr. Anton Berns at The Netherlands Cancer Institute for the generous gift of p53 conditional mutant mice. This work was funded in part by the following: National Institutes of Health R01 CA107469 (to C.G.K.); Cancer Prevention & Research Institutes of Texas (RP160710 to M.-C.H.); Breast Cancer Research Foundation (BCRF-17-069 to M.-C.H. and G.N.H). National Breast Cancer Foundation, Inc.; Patel Memorial Breast Cancer Endowment Fund; The University of Texas MD Anderson-China Medical University and Hospital Sister Institution Fund (to M.-C.H.); NIH T32 Training Grant in Cancer Biology (5T32CA186892, to L.-C.C.); CPRIT Research Training Program (RP101502, 140106, and 170067; to H.-W.K.); and the Ministry of Science and Technology (Taiwan) New Partnership Program for the Connection to the Top Labs in the World (Dragon Gate Program; 107-2911-I-006-519 to Y.-Y.C.).

## Author contributions

L.N. and Y.W. designed and performed the experiments, analyzed data, and wrote the manuscript; F.Z., Y.-H.H., L.-C.C., W.X., B.K., C.Z., R.D., J.T., J.Y., Y.-Y.C., X.Z., Y.H., J.H., L.H., H.-W.K., W.L., H.Y., J.-M.H., Y.Y., D.N.P. performed experiments and analyzed data; J.L.H. provided scientific input and wrote the manuscript; C.G.K. provided EZH2^WT transgenic mouse, N.E.D. and G.N.H. provided clinical input; M.-C.H. supervised the entire project, designed the experiments, analyzed data, and wrote the manuscript.

## Competing interests

The authors declare no competing interests.

## Additional information

Lei Nie[1,15], Yongkun Wei[1,15], Fei Zhang[1,2], Yi-Hsin Hsu[1], Li-Chuan Chan[1,3], Weiya Xia[1], Baozhen Ke[1], Cihui Zhu[1], Rong Deng[1,4], Jun Tang[1,5], Jun Yao[1], Yu-Yi Chu[1], Xixi Zhao[1,6], Ye Han[1,7], Junwei Hou[1], Longfei Huo[1], How-Wen Ko[1,3], Wan-Chi Lin[1], Hirohito Yamaguchi[1,8], Jung-Mao Hsu[1], Yi Yang[1], Dean N. Pan[1,9], Jennifer L. Hsu[1,10,11], Celina G. Kleer[12], Nancy E. Davidson[13], Gabriel N. Hortobagyi [14] & Mien-Chie Hung[1,3,10,11]

[1]Department of Molecular and Cellular Oncology, The University of Texas MD Anderson Cancer Center, Houston, TX 77030, USA. [2]Department of General Surgery, Xinhua Hospital affiliated to Shanghai Jiao Tong University School of Medicine, Shanghai 20009, China. [3]Graduate School of Biomedical Sciences, The University of Texas Health Science Center at Houston, Houston, TX 77030, USA. [4]State Key Laboratory of Oncology in South China, Cancer Center, Sun Yat-Sen University, Guangzhou, China. [5]Department of Breast Oncology, Cancer Center, Sun Yat-Sen University,

Guangzhou, China. [6]Department of Oncology, The Second Affiliated Hospital of Xi'an Jiaotong University, Shaanxi, China. [7]Department of Breast Surgery, Shengjing Hospital of China Medical University, Shenyyang 110003, China. [8]Cancer Research Center, Qatar Biomedical Research Institute, Hamad Bin Khalifa University, Qatar Foundation, PO Box 34110 Doha, Qatar. [9]College of Liberal Arts, The University of Texas at Austin, Austin, TX 78712, USA. [10]Graduate Institute of Biomedical Sciences and Center for Molecular Medicine, and Office of the President, China Medical University, Taichung 404, Taiwan. [11]Department of Biotechnology, Asia University, Taichung 413, Taiwan. [12]Department of Pathology, University of Michigan Medical School, Ann Arbor, MI 48109, USA. [13]Fred Hutchinson Cancer Research Center, Seattle, WA 98109, USA. [14]Department of Breast Medical Oncology, The University of Texas MD Anderson Cancer Center Houston, Houston, TX 77030, USA. [15]These authors contributed equally: Lei Nie and Yongkun Wei.

