## [Peer Review File with Redaction · Nature Communications]

Reviewers' comments:

Reviewer #1 (Remarks to the Author):

It is widely held that transitioning triple negative breast cancers (TNBC) into those that are "luminal" and dependent on estrogen receptor- α (ER) signaling would provide an avenue to treat this highly aggressive disease. However, approaches to accomplish this have thus far been unsuccessful given the limited ability to activate ER dependency. This report builds on the authors' prior discoveries that EZH2 is overexpressed in triple negative breast cancer (TNBC), can be phosphorylated at T416 by CDK2, and that high pT416-EZH2 correlates with poor survival. The manuscript focuses on determining whether the CDK2/pT416-EZH2 signaling axis drives TNBC and if its inhibition can promote luminal phenotypes that are then susceptible to estrogen receptor-targeted therapies. As an initial assessment of the role of phosphorylated EZH2 in TNBC, the authors generated a novel transgenic mouse model that overexpresses a phospho-mimicking EZH2 (EZH2T416D) driven by the MMTV promoter (Tg-EZH2T416D). These mice develop tumors that express basal markers (CK14, SMA). Furthermore, crossing the Tg-EZH2T416D mice to a luminal tumor model (Tg-Neu), results in progeny whose tumors now have basal-like characteristics (greater incidence of metastasis, expression of basal markers, decreased survival). A combination of in vitro and in vivo (xenograft studies) approaches revealed that blocking EZH2 or CDK2 with selective inhibitors (EPZ-6438/GSK342 and dinaciclib, respectively) results in the upregulation of ER expression and decreased tumorigenesis. ChIP assays were used to show that EZH2 directly regulates the ER encoding gene with EZH2 directly binding to the ESR1 promoter and this binding was associated with increased H3K27Me3. To therapeutically capitalize on these findings, the authors test whether EZH2 or CDK2 inhibitors can induce ER expression and whether they are synergistic with the selective estrogen receptor modulator (SERM), tamoxifen. The combination treatment modestly inhibits tumor growth to an extent greater than either drug alone and may improve survival. The authors conclude that combined treatment with an EZH2 or CDK2 and tamoxifen may provide an important therapeutic avenue for treating TNBC.

Major strengths of the paper include the novel transgenic mouse for TNBC and the demonstration that phosphorylated EZH2 contributes to the basal phenotype of breast cancers. In addition, it focuses on a major clinical need and the concept of switching cellular phenotypes in TNBC to those that are treatable with the current standard of care for luminal tumors. However, there are several limitations to the study that preclude some of the conclusions of the authors. Most importantly, the authors repeatedly indicate that the phosphorylation of EZH2 by CDK2 is responsible for the TNBC phenotype. However, they have not directly shown that the primary effect of CDK2 inhibition is restricted to altering EZH2 and its transcriptional targets. CDK2 is a major kinase that modulates many proteins that extend beyond EZH2. Demonstrating that CDK2 inhibition changes the phosphorylation of EZH2 and reduces the expression of a single target (ESR1) does not rule out the likely possibility that other targets of CDK2 are also involved in mediating the effects of the inhibitors. This would require add-back experiments with phospho-mimic EZH2 to demonstrate that the CDK2 inhibitors are no longer effective. In addition to this limitation, there is a major concern regarding the up-regulation of ER upon treatment with CDK2 or EZH2 inhibitors. Most of the western blots shown indicate that there is basal ER expression in the many TNBC models used as well as in the Neu mouse model. However, it is well established that these cell lines/mouse model do not express ER. This suggests that either the antibody used for ER is not selective or that the exposures of the blots are excessive for detecting a modicum of ER expression. Lastly, the effects of tamoxifen when added to CDK2 or EZH2 inhibitors are quite modest in both the 3D cultures as well as the mouse models. Thus, the implication that combining tamoxifen with a CDK2 or EZH2 inhibitor may be an effective treatment for patients with a highly aggressive disease is overstated.

Additional major concerns are enumerated below:

1. The authors address a key point regarding potential toxicity of the combined drugs and reduce

the dose for the last study presented in the paper. This raises the important question of whether the experiments in Figure 4 utilized levels of combined drugs that are toxic. If so, these data are potentially confounded by toxicity. To obviate this concern, minimal evidence that the combination is not toxic at the doses used in Figure 4 are necessary.

2. The report of a CI value within the mouse models requires more rigorous assessments at multiple doses.
3. The mouse survival study in Figure 4G was too short to conclude a substantial increase in survival and the study in Figure 5B is unacceptably underpowered.
4. To conclude that treatment with EZH2 or CDK2 inhibitors truly makes the tumors dependent upon ER, the selective estrogen receptor degrader (SERD), fulvestrant, should also be assessed.
5. There are multiple examples where no statistical analysis appears to have been conducted, yet the authors conclude that significant differences are observed between groups (Fig. 3A, 4A).

Minor Concerns:

1. It is unclear how many times each western blot was repeated and there is no quantitation of these blots.
2. There are no error bars in Figure 2A.
3. The concentration of tamoxifen used in the in vitro models is not stated.
4. Animal numbers should be reported for each experiment.
5. Is Figure 1D overall survival or tumor-free survival?
6. The images in Figure 1A are too small to visualize the metastatic lesions within the lung.
7. Figure 1: What percentage of Tg-Neu;EZH2 t416D mice were CK18-positive?
8. Figure S2C: T416D transduced cells still have equal levels of GATA3 and CK18 expression to that of the vector, yet the TgEZH2T416D mice displayed decreased levels. This discrepancy should be addressed.
9. In Figure 5B, the tamoxifen (green line) is missing from the graph.

Reviewer #2 (Remarks to the Author):

This is an extraordinary manuscript, describing an activation of EZH2 that confers triple negative mammary cancer in mice. There are no molecular therapeutics for TNBC and any evidence of a common mechanistic pathway is of both basic and translational importance. Crossing of this line into a HER2 GEM resulted in conversion of a HER2+ model to TNBC. This phenotypic plasticity has been reported in metastatic disease and renders HER2 therapies inactive, so it is important to understand. Crossing of a conditional knockout of EZH2 to a BRCA GEM induced ER expression, another potentially translatable finding. Using xenograft models, preclinical testing of CDK2 and EZH2 inhibitors provided proof of principle that TNBC could be converted to an ER+ phenotype, sensitive to tamoxifen. I have only a few major questions and a number of minor, explanatory requests.

1. In Fig 1D, is the survival of the crosses different from Neu alone?
2. For the 4T1 model, metastatic ability should be shown throughout, in addition to primary tumor size data and survival. The survival is generally more impressive than primary tumor growth, and this could be that primaries reach a size where euthanasia is required faster, or metastatic disease limits lifespan.
3. The authors concentrated on expression of therapeutically important markers such as ER. However, activation of EZH2 likely influences a number of gene expression patterns and I would like to see some global analysis of these tumors. I think figures can be condensed to permit this.

Minor:

1. Introduction, please give numbers here for EZH2 overexpression in breast cancer from TCGA. Also co-expression with cyclin E.
2. Please give some background and references on EPZ6438. Specificity, potency? Same for GSK343. Same for CDK2i.

Point-by-Point Response to Reviewers

Reviewer #1's Comments: *It is widely held that transitioning triple negative breast cancers (TNBC) into those that are "luminal" and dependent on estrogen receptor-alpha (ER) signaling would provide an avenue to treat this highly aggressive disease. However, approaches to accomplish this have thus far been unsuccessful given the limited ability to activate ER dependency. This report builds on the authors' prior discoveries that EZH2 is overexpressed in triple negative breast cancer (TNBC), can be phosphorylated at T416 by CDK2, and that high pT416-EZH2 correlates with poor survival. The manuscript focuses on determining whether the CDK2/pT416-EZH2 signaling axis drives TNBC and if its inhibition can promote luminal phenotypes that are then susceptible to estrogen receptor-targeted therapies. As an initial assessment of the role of phosphorylated EZH2 in TNBC, the authors generated a novel transgenic mouse model that overexpresses a phospho-mimicking EZH2 (EZH2T416D) driven by the MMTV promoter (Tg-EZH2T416D). These mice develop tumors that express basal markers (CK14, SMA). Furthermore, crossing the Tg-EZH2T416D mice to a luminal tumor model (Tg-Neu), results in progeny whose tumors now have basal-like characteristics (greater incidence of metastasis, expression of basal markers, decreased survival). A combination of in vitro and in vivo (xenograft studies) approaches revealed that blocking EZH2 or CDK2 with selective inhibitors (EPZ-6438/GSK342 and dinaciclib, respectively) results in the upregulation of ER expression and decreased tumorigenesis. ChIP assays were used to show that EZH2 directly regulates the ER encoding gene with EZH2 directly binding to the ESR1 promoter and this binding was associated with increased H3K27Me3. To therapeutically capitalize on these findings, the authors test whether EZH2 or CDK2 inhibitors can induce ER expression and whether they are synergistic with the selective estrogen receptor modulator (SERM), tamoxifen. The combination treatment modestly inhibits tumor growth to an extent greater than either drug alone and may improve survival. The authors conclude that combined treatment with an EZH2 or CDK2 and tamoxifen may provide an important therapeutic avenue for treating TNBC.*

Major strengths of the paper include the novel transgenic mouse for TNBC and the demonstration that phosphorylated EZH2 contributes to the basal phenotype of breast cancers. In addition, it focuses on a major clinical need and the concept of switching cellular phenotypes in TNBC to those that are treatable with the current standard of care for luminal tumors. However, there are several limitations to the study that preclude some of the conclusions of the authors. Most importantly, the authors repeatedly indicate that the phosphorylation of EZH2 by CDK2 is responsible for the TNBC phenotype. However, they have not directly shown that the primary effect of CDK2 inhibition is restricted to altering EZH2 and its transcriptional targets. CDK2 is a major kinase that modulates many proteins that extend beyond EZH2. Demonstrating that CDK2 inhibition changes the phosphorylation of EZH2 and reduces the expression of a single target (ESR1) does not rule out the likely possibility that other targets of CDK2 are also involved in mediating the effects of the inhibitors. This would require add-back experiments with phospho-mimic EZH2 to demonstrate that the CDK2 inhibitors are no longer effective. In addition to this limitation, there is a major concern regarding the up-regulation of ER upon treatment with CDK2 or EZH2 inhibitors. Most of the western blots shown indicate that there is basal ER expression in the many TNBC models used as well as in the Neu mouse model. However, it is

well established that these cell lines/mouse model do not express ER. This suggests that either the antibody used for ER is not selective or that the exposures of the blots are excessive for detecting a modicum of ER expression. Lastly, the effects of tamoxifen when added to CDK2 or EZH2 inhibitors are quite modest in both the 3D cultures as well as the mouse models. Thus, the implication that combining tamoxifen with a CDK2 or EZH2 inhibitor may be an effective treatment for patients with a highly aggressive disease is overstated.

Additional major concerns are enumerated below:

- 1. The authors address a key point regarding potential toxicity of the combined drugs and reduce the dose for the last study presented in the paper. This raises the important question of whether the experiments in Figure 4 utilized levels of combined drugs that are toxic. If so, these data are potentially confounded by toxicity. To obviate this concern, minimal evidence that the combination is not toxic at the doses used in Figure 4 are necessary.*
- 2. The report of a CI value within the mouse models requires more rigorous assessments at multiple doses.*
- 3. The mouse survival study in Figure 4G was too short to conclude a substantial increase in survival and the study in Figure 5B is unacceptably underpowered.*
- 4. To conclude that treatment with EZH2 or CDK2 inhibitors truly makes the tumors dependent upon ER, the selective estrogen receptor degrader (SERD), fulvestrant, should also be assessed.*
- 5. There are multiple examples where no statistical analysis appears to have been conducted, yet the authors conclude that significant differences are observed between groups (Fig. 3A, 4A.)*

Minor Concerns:

- 1. It is unclear how many times each western blot was repeated and there is no quantitation of these blots.*
- 2. There are no error bars in Figure 2A.3. The concentration of tamoxifen used in the in vitro models is not stated*
- 4. Animal numbers should be reported for each experiment.*
- 5. Is Figure 1D overall survival or tumor-free survival?*
- 6. The images in Figure 1A are too small to visualize the metastatic lesions within the lung.*
- 7. Figure 1: What percentage of Tg-Neu;EZH2 t416D mice were CK18-positive.*
- 8. Figure S2C: T416D transduced cells still have equal levels of GATA3 and CK18 expression to that of the vector, yet the TgEZH2T416D mice displayed decreased levels. This discrepancy should be addressed.*
- 9. In Figure 5B, the tamoxifen (green line) is missing from the graph.*

Authors' Response: We deeply appreciate the reviewer for the comments and suggestions to improve the scientific merit of the manuscript. Below please find our response to each of these comments.

Reviewer's Comment #1: *Demonstrating that CDK2 inhibition changes the phosphorylation of EZH2 and reduces the expression of a single target (ESR1) does not rule out the likely possibility that other targets of CDK2 are also involved in mediating the effects of the inhibitors. This would*

require add-back experiments with phospho-mimic EZH2 to demonstrate that the CDK2 inhibitors are no longer effective

Authors' Response: We thank the reviewer for the suggestion and performed a new experiment in TNBC cell line BT-549 to address this issue. To demonstrate that CDK2 inhibition blocks the phosphorylation of EZH2 at T416 and thereby reduces the expression of targeted gene ESR1, we performed an add-back experiment in TNBC cell line BT-549 as suggested by the reviewer. We established a control and a myc-tagged phospho-mimic EZH2 (myc-EZH2^{T416D}) stable cell line by lentiviral infection (Fig. S3d, lane 4 and 5) and treated them with 50 nM CDK2 inhibitor (Dinaciclib, DINA) for 72 hrs. Western blot analysis showed that ectopic expression of phosphomimic myc-EZH2^{T416D} in BT-549 cells indeed abolished CDK2i-induced ER α expression (Fig. S3d, lane 3 vs. 5). This suggested that the effect of CDK2 inhibitor on ER α re-expression is dependent on the phosphorylation of EZH2, and T416 is the major phosphorylation site of EZH2 responsible for suppressing ER α expression. We thank the reviewer for the suggested add-back experiments which indeed strengthen the conclusion that site-specific phosphorylation of EZH2 by CDK2 contributes to the inhibition of ER α expression. We added a description above on pages 9 of the revised manuscript.

Fig. S3d. Ectopic expression of EZH2^{T416D} in TNBC cells abolishes the reactivation of ER α gene expression induced by CDK2 inhibitor (CDK2i, dinaciclib). Add-back experiment of CDK2i-resistant EZH2^{T416D} mutant to BT-549 cells was performed by lentiviral infection. The cells were treated with dinaciclib at 50 nM for 72 hours. The cells expressing EZH2^{T416D} failed to respond to CDK2i treatment (Lane 3 vs 5). The luminal cell lysate of T47D (lane 1) was used for ER α positive control.

Reviewer's Comment #2: . In addition to this limitation, there is a major concern regarding the up-regulation of ER upon treatment with CDK2 or EZH2 inhibitors. Most of the western blots shown indicate that there is basal ER expression in the many TNBC models used as well as in the Neu mouse model. However, it is well established that these cell lines/mouse model do not express ER. This suggests that either the antibody used for ER is not selective or that the exposures of the blots are excessive for detecting a modicum of ER expression.

Authors' Response: We thank the reviewer for pointing this out. As the reviewer mentioned, in general, TNBCs are ER α negative. Pathologically, less than 1% ER positive cells in tumor sections by IHC staining are defined as negative. TNBCs are highly heterogeneous, and study has shown

that some TNBC cell lines express ER α at low levels (Ford CH et al. *Anticancer Res.* 31(2):521-7, 2011). We agree with the reviewer that Tg-Neu mammary tumors are identified as luminal but ER negative (Pfefferle AD et al. *Genome Biol.* 14:R125, 2013). The literatures demonstrated that ER α gene expression is retained at the premalignant stages of Tg-Neu mice, and ER α protein is lost after tumor progression (Zhang X. et al. *Oncogene*, 24:4220-4231; Rossi EL et al. *Cancer Res.* 77:2500-2511, 2017; Ma Z et al. *Carcinogenesis*, 39 (10):1264-1273). To further verify whether the ER α protein is detectable at the late stage of the tumors, we examined more tumor tissues from the Tg-Neu mice. As shown in the figure S3g, compared with T47D luminal breast cancer cells ER α protein in a few tumor samples from the Tg-Neu mice (Tumor #1 and 4) were very low but detectable by western blot. This result suggests that the low level of ER α protein in tumor samples taken from the Tg-Neu mice possibly contain premalignant components of the mammary glands.

Fig. S3g. Protein levels of GATA3 and ER α in the mammary gland tumors from Tg-Neu mice. ER α , GATA3 and transgene Her2/Neu proteins in whole lysates of six primary tumor tissues were immunoblotted with indicated antibodies. The luminal breast cancer cell line T47D was used for positive control.

To further evaluate the specificity of the ER α antibody (Abcam, Cat#108398) used in our experiments, we generated ER α knockout T47D cell line by Cas9/CRISPR-sgRNA against ER α and treatment of T47D cells with ER antagonist fulvestrant. Western blot analysis showed that knockout of ER α in T47D cells abolished the ER α signal (Fig. S3h, lane 1 vs 2). In BT-549 TNBC cells, the level of ER α was significantly induced by 50 nM CDK2i treatment (72 h) compared with the vehicle control (Fig. S3h, lane 3 vs. 4). Importantly, the induced ER α was detected at the same molecular weight as that in luminal T47D cells. In addition, in T47D luminal breast cancer cell line, the ER α protein level was decreased with increasing concentrations of fulvestrant (Fig. S3i). Notably, the expression level of ER α induced by EZH2i and CDK2i in human TNBC cells was comparable to that observed in the T47D luminal cell line (Fig. S3a). These findings indicated that the antibody against ER α we used in the experiments is ER α specific.

Fig S3h. Anti-ERα antibody specifically recognized ERα in ERα⁺ T47D cells and CDK2i-induced ERα in BT-549 cells. Cas-9/sgRNA targeting deletion of ERα in T47D cells abolished ERα signal but CDK2i induced ERα expression in TNBC BT-549 cells in western blot analysis.

Fig. S3i. ERα in the luminal cell line T47D is decreased with increased fulvestrant. ERα protein in the lysates of T47D treated with vehicle or fulvestrant was immunoblotted with polyclonal anti-ERα antibody from Abcam.

Fig. S3a Induction of ERα expression by EZH2i in TNBCs is dose-dependent. The TNBC cells were treated with EZH2i GSK343 at different concentrations for 72 hours. Whole cell lysates were used for immunoblot analysis. Dinaciclib (DINA) treatment and T47D cell lysate were used for ERα positive control.

Reviewer's Comment #2: *The effects of tamoxifen when added to CDK2 or EZH2 inhibitors are quite modest in both the 3D cultures as well as the mouse models. Thus, the implication that combining tamoxifen with a CDK2 or EZH2 inhibitor may be an effective treatment for patients with a highly aggressive disease is overstated.*

Authors' Response Although the inhibitory effect of the combination regimen on xenograft tumor growth was not impressive for EZH2i, statistical analysis indicated that a significant difference between EZH2i and control groups in the SUM-149 xenograft model ($p < 0.01$, Fig. 4e). EZH2i has very short half-life, and thus the administration of the inhibitor to effectively suppress tumor growth in vivo were twice or three times daily at a dose of 80.5–322mg/kg in

the previous study (Knutson SK et al. Mol Cancer Ther; 13:842-854). Due to limitation of animal protocol for oral feeding and efficacy assessment of the combination therapy of EZH2i and tamoxifen, we treated tumor-bearing mice with EZH2i at lower dose (≤ 100 mg/kg) once daily (Fig. 4e). Indeed, our *in vitro* and *in vivo* experiments showed that combination treatment of EZH2i or CDK2i and tamoxifen in TNBC resulted in significant inhibition of tumor growth ($p < 0.01$ and $p < 0.001$, respectively, Fig. 4e and f) even though they could not fully suppress the tumor growth at the dosage administered. However, we observed a meaningful benefit of the combination therapy in the survival of the tumor-bearing mice (Fig. 4g). TNBC has poorest outcome among all breast cancer subtypes and has a median overall survival of 10.2 months with current therapies. Thus, it is worth exploring such alternative way to improve survival rate of the TNBC patients, particularly for patients with disease relapse and chemotherapy-resistance for which no drugs are currently available. It would be worthwhile to optimize the dose, frequency, duration, and route of administration of the CDK2i and EZH2i combined with either tamoxifen or fulvestrant so to improve the tumor inhibitory effects and survival of TNBC tumor-bearing mice. The EZH2 and CDK2 inhibitors used in our experiments are all currently in clinical trials, and the results are encouraging. Therefore, the combination therapy was investigated as an alternative treatment for TNBCs.

Fig. 4e and f. Combination therapy of EZH2i and CDK2i with tamoxifen (TAM) in SUM-149 xenograft mouse model significantly inhibit tumor growth. TNBC SUM-149 cells were orthotopically inoculated on nude mice. The tumor bearing mice were treated with EZH2i (GSK343) and CDK2i (dinaciclib) alone at a dose of 100 mg/kg and 8 mg/kg, respectively, or combined with tamoxifen at 4 mg/kg. The tumor burdens were measured with digital caliper. *not significant, *** $p < 0.01$, **** $p < 0.001$. $n = 10$.

Fig. 4g. Combination therapy of CDK2i with tamoxifen improves survival of TNBC tumor-bearing Mice. TNBC SUM-149 cells orthotopically injected into mammary gland pads of Nude/Nude female mice. When the tumors reached to 70-90mm³ in size, the tumor-bearing mice were treated with CDK2i (Dinaciclib) at 8 mg/kg alone or combined with tamoxifen at 4 mg/kg daily. Survival curves of tumor-bearing mice treated with CDK2i, TAM, or their combination. The survival rates plotted and analyzed with GraphPad Prism 7. Log-Rank test, *not significant, *** p < 0.01, **** p < 0.001. n = 10.

Other Major Concerns:

Point #1: *The authors address a key point regarding potential toxicity of the combined drugs and reduce the dose for the last study presented in the paper. This raises the important question of whether the experiments in Figure 4 utilized levels of combined drugs that are toxic. If so, these data are potentially confounded by toxicity. To obviate this concern, minimal evidence that the combination is not toxic at the doses used in Figure 4 are necessary.*

Authors' Response: We greatly appreciate you asking a critical question about the toxicity at high doses of CDK2i for longer period of time (21-day treatment schedule). We carried out one additional experiment to test whether a high dose of CDK2 inhibitor we used in the in vivo assay induces toxic effects. DINA (8 mg/kg) combined with tamoxifen (4 mg/kg) as described in Figure 4 was administered in five immunocompetent Balb/c mice daily for 21 days and serum was collected for biochemical analysis (Fig. S5 c). The mice body weight was monitored for 25 days (see Fig. S5 d). Similar to the 15-day treatment schedule (Fig. S5a and b), the routine biochemical indicators of kidney and liver function were within the normal range with no substantial changes in body weight in the treated mice (Fig. S5c and d), suggesting the absence of cytotoxic effects. We have incorporated these data into Fig. S5 in the revised manuscript.

Figure S5. Toxicity of combination therapy of the CDK2i or EZH2i and TAM on TNBC in vivo. (a and b) Changes in liver and kidney function indicators, and body weight of mice from Figure 4 (c and d) after administration of EZH2i (EPZ-6438; 34 mg/kg) or CDK2i (DINA; 8 mg/kg) alone or in combination with tamoxifen (4 mg/kg) for 15 days. (c and d) Changes in liver and kidney function indicators, and body weight of mice administered the combination (+) of CDK2i (8 mg/kg) and tamoxifen (4 mg/kg) for 21 days compared with those without (–) treatment (n = 5). Dashed lines indicate the normal range. AST, aspartate aminotransferase. ALT, alanine aminotransferase. BUN, Blood urea nitrogen.

Point #2: *The report of a CI value within the mouse models requires more rigorous assessments at multiple doses.*

Authors' Response: We thank the reviewer for the comment. Both EZH2 and CDK2 inhibitors combined with tamoxifen or fulvestrant exhibited synergistic effects against TNBC growth in vitro (Fig. S4a and b). We agree with the reviewer that determining the precise CI value within the mouse model requires multiple doses, and a large number of mice. However, it may be impractical to perform such rigorous assessments at this stage. Therefore, we deleted the CI index from the animal experiment data in the revised manuscript.

Fig. S4 Synergistic inhibitory effect of CDK2i and EZH2i with hormonal therapies on TNBCs. TNBC cells were pretreated with CDK2i (DINA; 10nM) or EZH2i (EPZ) at 12.5 μ M for 3 days and seeded in 96-well plate followed by treatment with or without tamoxifen or fulvestrant for 72 h. Combination Index (CI) values were calculated as previously described by Chou T-C, (Chou, Ting-Chao. *Cancer Res.* 70:440-6, 2010). Criterion for combination index (CI): Synergy, $0.3 < CI \leq 0.85$; Strong synergy, $CI \leq 0.3$. TAM, Tamoxifen; Ful, fulvestrant.

Point #3: *The mouse survival study in Figure 4G was too short to conclude a substantial increase in survival and the study in Figure 5B is unacceptably underpowered.*

Authors' Response: We thank the reviewer for the comments. We presented longer observation data for survival (58 days). We agree with you that our observation period (about two months) was not long enough to draw a conclusion with “substantial”. The “substantial” conclusion in our manuscript is based on the statistical analysis, $p < 0.001$ (Fig. 4g). We have changed our wording to “significant” in the revised manuscript (page 13). For Figure 5b, we agree that it would be more convincing if the mouse number is greater than 5. Statistically, for the final tumor size assessment, it is anticipated that there are about 5 mice in each experiment group based on Sample Size Calculations (IACUC). To detect an effect size of 2 groups, using a two-sided t-test with a significance level of 0.05, five mice would achieve 90 % power (also please see p values in the figures). We agree with the reviewer that to reach statistical significance in the experimental condition due to unexpected variation of mice more than 5 mice will be ideal and more convincing. Because our previous Figure 5b did not present a clear cut conclusion, we have moved this figure to the supplement as a supporting data (Fig. S6).

Fig. 4g and Fig. S6b. Combination therapy of CDK2i with tamoxifen improves survival of TNBC tumor-bearing mice. TNBC cells were orthotopically injected into mammary gland pads of Nude/Nude female mice. When the tumors reached to 70–90 mm³ in size, the tumor-bearing mice were pre-treated with CDK2i (DINA) at 8 mg/kg (Fig. 4g, n = 10) or 2 mg/kg (Fig. S6b, n = 5) for 3 days followed by combination treatment of CDK2i and tamoxifen (4 mg/kg) or DINA alone. The survival rates were plotted and analyzed with GraphPad Prism 7. Log-Rank test, *not significant, ** p < 0.05, *** p < 0.01, **** p < 0.001.

Point #4: To conclude that treatment with EZH2 or CDK2 inhibitors truly makes the tumors dependent upon ER, the selective estrogen receptor degrader (SERD), fulvestrant, should also be assessed.

Authors' Response: We greatly appreciate the reviewer's suggestion. Accordingly, we performed in vitro viability assay to assess the effects of fulvestrant on TNBC cell lines MDA-

MB-231. Similar to tamoxifen, we observed strong synergistic inhibitory effect ($CI \leq 0.3$) on the TNBC cell proliferation by the combination treatment of CDK2i and EZH2i with fulvestrant (Fig. S4b). Our results showed fulvestrant (Ful) alone did not induce significant anti-proliferative effects but when combined with EZH2i and CDK2i, we observed significantly reduced the cell viability compared with either CDK2i or EZH2i treatment alone (Fig. S4c, upper panel). Moreover, Western blot analysis indicated that pretreatment of the MDA-MB231 cells with either EZH2i or CDK2i alone induced an apparent amount of ER α compared with the luminal cell line T47D (Fig. S4c, bottom, lane 5 and 8). The combined treatment of CDK2i or EZH2i and fulvestrant (Ful) abolished the inhibitors-induced ER α expression in TNBC cells (Fig. S4c, bottom panel, lane 6 and 9), indicating that the inhibitor-induced ER α proteins were degraded by fulvestrant. These results suggested that CDK2i and EZH2i induce ER α expression in TNBC cells, rendering them sensitive to estrogen receptor degrader fulvestrant.

Fig. S4b Synergistic inhibitory effect of CDK2i and EZH2i with hormonal therapies on TNBCs. TNBC cells were pretreated with CDK2i (DINA; 10nM) or EZH2i (EPZ) at 12.5 μ M for 3 days and seeded in 96-well plate followed by treatment with or without fulvestrant for 72 h. Combination Index (CI) values were calculated as previously described (Chou TC, 2010). Criterion for combination index (CI): Synergy, $0.3 < CI \leq 0.85$; Strong synergy, $CI \leq 0.3$. Ful, fulvestrant.

Figure S4c. Synergistic inhibitory effect of CDK2i and EZH2i combined with hormone therapy.

Cell viability of MDA-MB231 cells treated with CDK2i (DINA) and EZH2i (EPZ) alone or combined with fulvestrant (upper panel). ER α protein levels in the treated with or without drugs were immunoblotted with specific antibodies (bottom panel). EPZ6438, EPZ; Dinaciclib, DINA; Ful, Fulvestrant. Student's T test, *not significant, ** $p < 0.05$, **** $p < 0.001$.

Point #5: *There are multiple examples where no statistical analysis appears to have been conducted, yet the authors conclude that significant differences are observed between groups (Fig. 3A, 4A.)*

Authors' Response: We apologize for this oversight. We have performed statistical analyses and we now added p values in those figures as shown below.

Fig. 3a. EZH2 directly targets the promoter of ER α gene and silences its expression. Under 3D culture conditions, MDA-MB231 cells were treated with DMSO, dinaciclib (DINA) or EPZ6438 (EPZ) for 3 days. The cells were collected to extract RNA. ER α expression was determined by real-time RT-PCR. Data are means \pm SD (n = 6). **p < 0.01, ***p < 0.001.

Figure 4. Functional reactivation of ER α expression in TNBCs by CDK2/EZH2 inhibitors creates an effective combination regimen with tamoxifen (TAM) for TN cancers *in vitro*. (a) Effective inhibition of TNBCs in 3D cultures by combination treatment of EZH2/CDK2 inhibitors (EPZ, GSK; SNS and DINA) and TAM determined by clonogenic assay. Representative images (upper panel) and bar graph of sphere numbers (bottom panel). Student's T test, *not significant, ** p < 0.05, **** p < 0.001.

Minor Concerns:

Minor Point #1: *It is unclear how many times each western blot was repeated and there is no quantitation of these blots.*

Authors' Response: We thank the reviewer for the comment. The representative Western blots shown were from at least two independent experiments. In the revised version, quantitation of the blots was performed by using ImageQuant TL Toolbox v8.1 software and normalized to the loading control, actin or tubulin, such as for Figure 4b. Quantitations are shown next to the blots in the revised manuscript.

Fig.2b. Treatment of 4T1 tumors in allograft mouse model with EZH2i induces GATA3 and ERα expression. Luminal cell, T47D breast cell line (T47D) ; Basal-like cell, T47D-EZH2^{T416D} tumor cells. The tumor-bearing mice were treated with EZH2i GSK343 at 100 mg/kg for 9 days. Protein levels in the lysates were immunoblotted with specific antibodies and quantitated with ImageQuant TL Toolbox v8.1 and normalized with actin.

Minor Point #2: *There are no error bars in Figure 2A.*

Authors' Response: We apologize for the missing error bars. We have added them in Figure 2a in the revised manuscript as shown below:

Fig.2a. Inhibition of EZH2 with specific inhibitor GSK343 reduces tumor burden in syngeneic mouse model. Murine basal-like mammary tumor cell line 4T1 cells were orthotopically inoculated into mammary fat pads of Balb/c female mice. The tumor-bearing mice were treated with either vehicle or EZH2i (GSK343) at 100 mg/kg for 9 days. Tumor volume was measured and presented in mean ± SD (n=6). ** p < 0.05.

Minor Point #3: *The concentration of tamoxifen used in the in vitro models is not stated.*

Authors' Response: We apologize for the oversight. We have added the tamoxifen concentration (2 μM) in Figure 4 (a and b) and the corresponding legend as shown below.

Figure 4a and b. Functional reactivation of ER α expression in TNBCs by CDK2/EZH2 inhibitors creates an effective combination regimen with tamoxifen (TAM) for TN cancers in vitro. (a)

Effective inhibition of TNBCs in 3D Matrigel cultures by combination treatments of CDK2/EZH2 inhibitors (EPZ, GSK; SNS and DINA) and Tamoxifen (TAM) at 2 μ M determined by clonogenic assay. Representative images of one experiment from three independent experiments is shown. (b) Combination treatment of the TNBCs in 3D culture with CDK2/EZH2 inhibitors and TAM induces apoptosis. An increased cleavage PARP protein in the combined treatment was detected by immunoblot. H3K27Me3 level was used as an index for inhibitory efficacy of the inhibitors.

*not significant, ** $p < 0.05$, *** $p < 0.001$.

Minor Point #4: Animal numbers should be reported for each experiment.

Authors' Response: We have included the animal numbers in the figures and/or figure legend.

Minor Point #5: Is Figure 1D overall survival or tumor-free survival?

Authors' Response: Figure 1D shows overall survival. We have indicated this in the main text and the corresponding figure legend in the revised manuscript.

Fig. 1d. Kaplan-Meier overall survival curves of Tg-EZH2^{WT}, Tg-Neu, Tg-EZH2^{T416D} and double transgenic Tg-EZH2^{WT};Neu and Tg-EZH2^{T416D};Neu mice. Long-Rank test, *not significant; *** p < 0.001; **** p < 0.0001.

Minor Point #6: The images in Figure 1A are too small to visualize the metastatic lesions within the lung.

Authors' Response: We have enlarged the images shown in Figure 1a.

Fig.1a. Representative photos of the mammary primary and metastatic tumors from TgMMTV-EZH2^{T416D} mouse. Primary tumor (upper panel); Metastatic tumor (Bottom). Arrows indicate the visible metastatic tumors in the lungs.

Minor Point #7: Figure 1: What percentage of Tg-Neu;EZH2 t416D mice were CK18-positive.

Authors' Response: We examined nine tumors from different Tg-Neu;EZH2T416D mice by IHC staining with results indicating that 22% of the Tg-Neu;EZH2T416D tumors were CK18-positive (among them, one tumor displayed CK18 and Ck14 double positive) and 78% were strong CK14-positive and CK18-negative. Taken together, more than 88% of the tumors from Tg-Neu;EZH2^{T416D} double transgenic mice displayed CK14 positive. These finding suggested that the EZH2^{T416D} expression in double-transgenic mice play a predominant role in tumor cell lineage commitment. We have ncorporated the observation into the revised manuscript (page 6).

Minor Point #8: Figure S2C: T416D transduced cells still have equal levels of GATA3 and CK18 expression to that of the vector, yet the TgEZH2T416D mice displayed decreased levels. This discrepancy should be addressed.

Authors' Response: We thank the reviewer for the comment. With longer culturing, we found that the levels of exogenous EZH2 protein in the stable cell lines were gradually reduced. Therefore, we repeated the experiment using newly established EZH2^{T416D} and vector control stable cell lines. Along with the newly established stable cells expressing EZH2^{T416D}, we probed both basal-like and luminal marker protein levels in the lysates of EZH2^{T416D} tumor and the tumor-derived cell lines. Compared with T47D vector control cell line, the ER α , GATA3 and CK18 protein levels in the EZH2^{T416D}-expressing stable cells, tumor tissue, and the tumor-derived cell line in 3 D Matrigel culture were significantly decreased. In contrast, basal-like marker CK14 was increased in all EZH2^{T416D}-expressing cells and tumor tissues (Fig.2c). These in vitro results are consistent with that shown in the tumors of Tg-EZH2^{T416D} mice. In addition, we observed that the basal-like T47D/EZH2^{T416D} cell line in 3D culture showed more significant increase of luminal markers ER α , GATA3, and CK18 than that in 2D culture after treatment of the tumor cells with EZH2 specific inhibitor EPZ-6438 (Fig. S2d). This may be due to the Matrigel-based 3D culture system can closely recapitulate cancer cell/matrix interaction and thus provide better microenvironment for the maintenance of tumor cell phenotype. We have incorporated current data into the revised manuscript as shown below.

Figure S2. Ectopic expression of EZH2^{T416D} but not T416A in T47D cells switches luminal cancer cells into basal-like type and enhances tumorigenesis independent of estrogen.

(c) Western blot analysis of luminal and basal-like markers in tumor-derived cell line and tumor tissue expressing EZH2^{T416D} in 3D Matrigel culture. (d) EZH2^{T416D} tumor-derived cell lines isolated from the tumors and selected by short trypsinization followed by G418 selection. EZH2^{T416D}-derived cells in 3D and 2D culture conditions were treated with EZH2 inhibitor EPZ6438 (EPZ) at 25 μ M for 72 h and subjected to Western blot analysis with the indicated antibodies. V, vector control cell line; T416D, EZHZ^{T416D}-expressing stable cell line. ER α , GATA3 and CK18 are human breast cancer luminal markers and CK14 basal-like marker.

Minor Point #9: In Figure 5B, the tamoxifen (green line) is missing from the graph.

Authors' Response: We apologize for this error and have corrected it in figure (see below) in the revised manuscript.

Fig S6b. Combination therapy of CDK2i at low dose with tamoxifen improves survival of TNBC tumor-bearing Mice. TNBC MDA-MB231 cells orthotopically injected into mammary gland pads of Nude/Nude female mice. When the tumors reached to 70-90 mm³ in size, the tumor-bearing mice were treated with CDK2i (Dinaciclib) at 2 mg/kg alone or combined with tamoxifen at 4 mg/kg daily for 4 weeks. The survival rates were plotted and analyzed with GraphPad Prism 7. Log-Rank test, *not significant, ** p < 0.05. n = 5.

Reviewer #2's Comments: *This is an extraordinary manuscript, describing an activation of EZH2 that confers triple negative mammary cancer in mice. There are no molecular therapeutics for TNBC and any evidence of a common mechanistic pathway is of both basic and translational importance. Crossing of this line into a HER2 GEM resulted in conversion of a HER2+ model to TNBC. This phenotypic plasticity has been reported in metastatic disease and renders HER2 therapies inactive, so it is important to understand. Crossing of a conditional knockout of EZH2 to a BRCA GEM induced ER expression, another potentially translatable finding. Using xenograft models, preclinical testing of CDK2 and EZH2 inhibitors provided proof of principle that TNBC could be converted to an ER+ phenotype, sensitive to tamoxifen. I have only a few major questions and a number of minor, explanatory requests.*

1. In Fig 1D, is the survival of the crosses different from Neu alone?
2. For the 4T1 model, metastatic ability should be shown throughout, in addition to primary tumor size data and survival. The survival is generally more impressive than primary tumor growth, and this could be that primaries reach a size where euthanasia is required faster, or metastatic disease limits lifespan.
3. The authors concentrated on expression of therapeutically important markers such as ER. However, activation of EZH2 likely influences a number of gene expression patterns and I would

like to see some global analysis of these tumors. I think figures can be condensed to permit this.

Minor:

1. Introduction, please give numbers here for EZH2 overexpression in breast cancer from TCGA. Also co-expression with cyclin E.

2. Please give some background and references on EPZ6438. Specificity, potency? Same for GSK343. Same for CDK2i.

Authors' Response: We thank the reviewer for the comments and suggestions. Below please find our response in a point-by-point manner.

Point #1: In Fig. 1D, Is the survival of crosses different from Neu alone?

Authors' Response: No, the survival of Tg-Neu and Tg-Neu;Tg-EZH2^{WT} were not significantly different according to statistical analysis ($p > 0.05$, Figure 1d). However, there is significantly difference between Tg-Neu and Tg-Neu; Tg-EZH2^{T416D} ($p < 0.0001$). The finding suggested that the expression of EZH2^{T416D} but not EZH2^{WT} in Tg-Neu mice contributes to the poorer survival than single transgenic mice.

Fig. 1d. Kaplan-Meier overall survival curves of Tg-EZH2^{WT}, Tg-Neu, Tg-EZH2^{T416D} and double transgenic Tg-EZH2^{WT};Neu and Tg-EZH2^{T416D};Neu mice. Long-Rank test, *not significant; *** $p < 0.001$; **** $p < 0.0001$.

Point #2: For the 4T1 model, metastatic ability should be shown throughout, in addition to primary tumor size data and survival. The survival is generally more impressive than primary tumor growth, and this could be that primaries reach a size where euthanasia is required faster, or metastatic disease limits lifespan.

Authors' Response: We thank the reviewer for the suggestion and added our observations on metastasis in Table 4 (see below). As compared with vehicle control and tamoxifen alone, inhibition of CDK2/EZH2 axis significantly reduced distant metastasis (lung) of 4T1 tumors, $p < 0.05$ (Table 4). EZH2i or CDK2i combined with tamoxifen further moderately reduced the lung

metastatic rate. The reduced lung metastasis in EZH2i and CDK2i treated group and combination group reflected the survival curves separated from the control and tamoxifen-treated groups (red line vs black and grey line) in Fig. 4g. As the reviewer mentioned that the survival rate is generally more impressive than suppression of tumor growth, our data indeed suggested that reduction of metastasis by blockade of CDK2/EZH2 signal axis may attribute at least partially to the improved survival of TNBC-bearing mice (Fig. 4g). The results are consistent with the reviewer's opinion.

Table 4. Inhibition of CDK2/EZH2 signaling axis reduces tumor lung metastasis.

Group	# Lung Metastasis	# Total Mice	Metastatic rate (%)	p value*
Vehicle	8	8	100	–
TAM	8	8	100	–
EPZ	4	8	50.0	0.0253
EPZ+TAM	3	8	37.5	0.0090
DINA	4	7	57.1	0.0454
DINA+TAM	4	8	50	0.0253

*Chi-Squared (χ^2) test, $p < 0.05$ is defined as statistically significant. EPZ, EPZ6438. DINA, Dinaciclib. TAM, tamoxifen.

Fig. 4g. Combination therapy of CDK2i with tamoxifen improves survival of TNBC tumor-bearing Mice. TNBC cells orthotopically injected into mammary gland pads of Nude/Nude female mice. When the tumors reached to 70-90mm³ in size, the tumor-bearing mice were treated with CDK2i (Dinaciclib) at 8 mg/kg alone or combined with tamoxifen at 4 mg/kg daily. Survival curves of tumor-bearing mice treated with CDK2i, TAM, or their combination (n = 10). Log-Rank test, *not significant, *** $p < 0.01$, **** $p < 0.001$.

Point #3: *The authors concentrated on expression of therapeutically important markers such as ER. However, activation of EZH2 likely influences a number of gene expression patterns and I would like to see some global analysis of these tumors. I think figures can be condensed to permit this.*

Authors' Response: We thank the reviewer for the excellent suggestion.

Following the reviewer's suggestion, we examined global transcriptome alternations of the TNBC tumors treated with CDK2i and EZH2i by RNAseq. We isolated total RNAs from the transplanted tumors of SUM-149 cell line treated with vehicle or CDK2i and EZH2i. Differential expression analysis on these tumors were performed by using Homo Sapiens (GRCH37/hg19) as reference genome and genome annotation. Setting fold change to 2 and $p < 0.05$ when using Limma package to call differential genes, we found that expression levels of more than 801 and 741 gene were altered by CDK2i and EZH2i treatment, respectively (RNAseq Data is available as request). Among differential changed genes induced by CDK2i and EZH2i, we defined top 108 common up- and down-regulated gene sets in the treated tumors (Fig. 5a and b). Enrichment analysis indicated that the up-regulated genes are mostly involved in (1) chemokine/cytokine pathway; (2) growth factor pathways and (3) Cell-cell/matrix interaction; Down-regulated genes were categorized into (1) RNA binding; (2) MHC pathways and (3) lipid metabolism (Fig. 5c). Interestingly, blockade of CDK2/EZH2 axis significantly up-regulated the gene expressions of chemokine/cytokine family members such as CXCL14, CXCL2, TNF α , IL1R2 etc., which are highly related to immune cell infiltration or migration in the tumor microenvironment, indicating that CDK2/EZH2 signaling axis is involved in the tumor immunity. In addition, among these common regulated genes, extracellular matrix proteases such as ADAM9 and ADAM19 are also the targeted genes of CDK2/EZH2 axis (Fig. 5 b and c). RNAseq results provided additional evidence to support our hypothesis that EZH2 is the epigenetic target gene of CDK2. Nevertheless, the biological functions of the common targeted genes of CDK2 and EZH2 remains to be further investigated. We have incorporated results and discussion into revised manuscript (pages 14–15 and pages 18–19, respectively. Please see new Figure 5 and Suppl. Data in Excel).

Figure 5. Global gene expression alterations in TNBC regulated by the CDK2/EZH2 signaling axis. The xenograft tumors of SUM-149 cells were treated with vehicle, CDK2i and EZH2i, respectively. Total RNAs were isolated from the tumors by using RNeasy® Mini Kit (Qiagen). Differential expression analysis on these tumors were performed by using Homo Sapiens (GRCH37/hg19) as reference genome and genome annotation. a, Venn diagram of the differential and common shared genes regulated by CDK2 and EZH2; b, Representative heat map of top regulated genes by blockade of CDK2/EZH2 signal axis; c, Representative top shared up- and down-regulated genes and their involved pathways. Enrichment analysis was performed as described by Kuleshov MV et al (Kuleshov MV. et al. Nucleic Acids Res. 70:W90-W97, 2016).

Minor Points:

Minor Point#1: Introduction, please give numbers here for EZH2 overexpression in breast cancer from TCGA. Also co-expression with cyclin E.

Authors' Response: Analysis of the TCGA dataset indicated overexpression of EZH2 in 94.5% of basal/TNBCs and in 81.9% of Her2-enriched subtypes. Consistently, overexpression of cyclin E genes (*CCNE1* and *CCNE2*) occurred at high percentages in basal-like and Her2-enriched subtypes of the breast cancers. Therefore, EZH2 and cyclin E genes (E1 and E2) are highly co-

expressed, 96.7% and 89.0%, respectively, in basal-like breast cancers. We have incorporated these analyses into the revised manuscript in Tables 1, 2, and 3 (also shown below).

Table 1. Overexpression of EZH2 in breast cancers.

EZH2 OE	Basal-like (n = 91)	HER2 (n = 116)	Luminal A (n = 130)	Luminal B (n = 172)	Normal-like (n = 6)
Yes	86	95	81	60	1
No	5	21	49	112	5
% OE	94.5	81.9	62.3	34.9	16.7

TCGA RNAseq Dataset; OE, overexpression.

Table 2. Overexpression of *CCNE1* and *CCNE2* in breast cancers.

CCNE1 OE	Basal-like (n = 91)	HER2 (n = 116)	Luminal A (n = 130)	Luminal B (n = 172)	Normal-like (n = 6)
Yes	88	94	52	53	1
No	3	22	78	119	5
% OE	96.7	81.0	40.0	30.8	16.7
CCNE2 OE	Basal-like (n = 91)	HER2 (n = 116)	Luminal A (n = 130)	Luminal B (n = 172)	Normal-like (n = 6)
Yes	81	112	109	110	1
No	10	4	21	62	5
% OE	89.0	96.6	83.8	64.0	16.7

TCGA RNAseq Dataset; OE, overexpression.

Table 3. Co-overexpression of EZH2 and Cyclin E genes in breast cancers.

EZH2/CCNE1	Basal-like (n = 91)	HER2 (n = 116)	Luminal A (n = 130)	Luminal B (n = 172)	Normal-like (n = 6)
Co-OE	84	82	43	33	1
% Co-OE	92.3	70.7	33.1	19.2	16.7

EZH2/CCNE2	Basal-like (n = 91)	HER2 (n = 116)	Luminal A (n = 130)	Luminal B (n = 172)	Normal-like (n = 6)
Co-OE	78	93	75	54	0
% Co-OE	85.7	80.2	57.7	31.4	0

TCGA RNAseq Dataset; OE, overexpression.

Minor Point #2: *Please give some background and references on EPZ6438. Specificity, potency? Same for GSK343. Same for CDK2i.*

Authors' Response: We thank the reviewer for the suggestion and added the following background and references on pages 3–4 of the revised manuscript as follows:

“Currently, two potent and selective S-adenosyl-methionine-competitive small molecule inhibitors of EZH2, GSK343 (GlaxoSmithKline) and EPZ-6438 (Epizyme), are under evaluation in phase I and II clinical trials¹⁴, with reports for the latter showing promising results¹⁵. EPZ-6438 selectively inhibits intracellular lysine 27 of histone H3 (H3K27) methylation in a concentration- and time-dependent manner in EZH2 wild-type and mutant lymphoma cells¹⁴. GSK343, a GSK126 derivative, has demonstrated activity against EZH2 in both enzymatic and cellular assays and has high selectivity for EZH2 over other methyltransferases, such as EZH1, which shares a 96%-sequence identity in the SET domain with EZH2¹⁶. The CDK2 inhibitor, SCH-727965 (Merck), also known as dinaciclib, is currently in phase III trials and selectively inhibits cyclin dependent kinases, including CDK1, CDK2, CDK5, and CDK9, with different IC₅₀¹⁷. In leukemia and several types of solid tumors, dinaciclib has demonstrated antitumor activity as a single agent. Importantly, dinaciclib has a safety profile in patients^{18, 19}.”

Reviewers' comments:

Reviewer #1 (Remarks to the Author):

The authors have adequately addressed most of this reviewers concerns, however, the following points should still be considered:

- 1) Western blots should be completed in triplicate, not duplicate as indicated by the authors.
- 2) The methods section describing the TCGA data should indicate the specific publication from which the data were obtained. In addition, the specific cut-off values for each gene should be indicated for tables 1 and 2.
- 3) ESR1 should be identified on the heat map for the RNA-seq data. In addition, the resolution of the heat map is too low to identify individual genes following zooming in on the image. The resolution should be corrected to allow identification of individual genes.
- 4) Rather than the RNA-seq data being "made available upon request", it should be submitted to a public repository.

Reviewer #3, Replacement Reviewer for Reviewer #2 (Remarks to the Author):

Transitioning between breast cancer subtypes during breast cancer development is an intriguing clinical observation. The understanding of the mechanisms that govern the TNBC phenotype has a clinical significance in guiding rationally designed treatment. In this manuscript by Hung et al. colleagues, the authors generated novel transgenic models (EZH2WT, EZH2T416D, and Tg-Neu;EZH2T416D) and demonstrated that hyperphosphorylation of T416 on EZH drives lineage fate of the basal-type breast cancer phenotype. The author further demonstrated that inhibition EZH2 or CDK2 using clinically available small molecule inhibitors reinitiated ERalpha expression in TNBCs and rendered TNBCs susceptible to anti-ER treatment. Overall, the data presented in this manuscript provide important mechanistic connection of EZH2-ERalpha axis in the vivo setting. Clinically, the implications of this finding could guide the future development of more effective combinatorial targeted therapies for TNBC patients. This revised manuscript sufficiently answered the issues raised by the previous reviewer. However, there are additional minor comments need to be addressed or clarified before warranting a publication at Nat Communications.

Major comments:

- 1) It is puzzling that T416A (phosphorylation-deficient) mutation showed no difference to EZH2 constructs on H3K27me3 (Fig. S2a). This argues against the dependency of 416 phosphorylation of EZH2's function in regulating ERalpha. This point should be discussed.
- 2) The Abcam ERalpha antibody (Cat#108398, as indicated in the author rebuttal letter) reacts with human ERalpha, not the mouse ERalpha, as indicated by Abcam. Although it is hard to exclude the possible cross-reaction without experimental evidence, the author should pay extra caution when using this antibody to claim mouse ERalpha expression. Moreover, the ERalpha antibody (Cat#108398) is a monoclonal antibody. It is confusing that the author stated in the main text line 238 that it is a polyclonal antibody and in the Figure legend of Fig. S3i. "... was immunoblotted with polyclonal anti-ER α antibody from Abcam."
- 3) The new Fig. S3d nicely demonstrated the T416 is a downstream target of CDK2i. It should be placed in the main Figure 2 instead of in the Supplementary Figures. Similarly, Fig. S4a and b are important data demonstrating the synergistic effects between two drugs. It needs to be in the main Figure 4.

- 4) In Table 4, it is not clear whether the vehicle group has always been used the reference (control) group when performing a chi-squared hypothesis testing. In addition, the author should indicate whether the p-value is generated from Fisher's exact test.
- 5) In the RNA-seq experiment, did authors observed an up-regulation of ESR1 (ER α) gene expression after EZH2i treatment?

Minor comments:

- 1) The color balance of CK14 and CK18 IHC staining (Fig. 1B, C) need to be improved. The high level of red background of the current picture makes very difficult to evaluate the results.
- 2) Since this is the first report of EZH2T416D transgenic mouse, more detailed description, e.g. genotyping primers and protocols, need to be provided.
- 3) Western blot validation of phosphor-CDK2 under Dinaciclib should be included in Fig. 2d for data consistency.
- 4) Fig 3e is not the true experimental result. It should be removed or be used as part of summary schematics.

Point-by-Point Response to Reviewer's Comments

Reviewer #1's Comments:

The authors have adequately addressed most of this reviewer's concerns, however, the following points should still be considered:

- 1) Western blots should be completed in triplicate, not duplicate as indicated by the authors.
- 2) The methods section describing the TCGA data should indicate the specific publication from which the data were obtained. In addition, the specific cut-off values for each gene should be indicated for tables 1 and 2.
- 3) ESR1 should be identified on the heat map for the RNA-seq data. In addition, the resolution of the heat map is too low to identify individual genes following zooming in on the image. The resolution should be corrected to allow identification of individual genes.
- 4) Rather than the RNA-seq data being "made available upon request", it should be submitted to a public repository.

Authors' Response: We deeply appreciate the reviewer for the comments and suggestions to improve the scientific merit of the manuscript. Below please find our response to each of these comments.

Point #1: Western blots should be completed in triplicate, not duplicate as indicated by the authors.

Response to Point #1: We agree with the reviewer's comment. Repeatability is crucial in research. Usually, Western blotting analysis run in triplicate will rule out experimental bias or some random error and thus help reduce variability and increase reproducibility. Indeed, the majority of our Western blots on CDK2i/EZH2i-induced targeted gene experiments were repeated by at least two investigators in triplicate, and a few of Western blotting analyses (e.g., Fig. S3a, d) were performed by two independent investigators in duplicate, which included dose-dependent effects (Fig. S3a) and time-course of the inhibitors (Fig. S3d) on the reactivation of ER α . In Figure 2b, the whole cell lysates were prepared from different tumors. In addition, we have corrected our labels in Figure 2b.

Figure 2. Pharmacological blockade of Cyclin E/CDK2-EZH2 signaling axis switches tumor cell lineage type from triple negative to ER α ⁺ luminal type in vivo and in vitro. (b) Whole lysates from different tumors treated with either vehicle (Veh1, Veh2) or GSK343 (GSK1, GSK2, and GSK3) were subjected to Western blot with the indicated antibodies. Cell lysates from luminal T47D cells and T47D-expressing

Point-by-Point Response to Reviewer's Comments

EZH2^{T416D} cells (basal-T416D) were added as positive and negative control, respectively, for GATA3 and ER α . Quantitation analysis of the images of experiments performed in triplicate was performed by using ImageQuant TL Toolbox v8.1 and normalized with loading control actin. The results were presented as bar graphs (right panel).

Point #2: *The methods section describing the TCGA data should indicate the specific publication from which the data were obtained. In addition, the specific cut-off values for each gene should be indicated for tables 1 and 2.*

Response to Point #2: We appreciate the reviewer's suggestion and cited appropriate publications for TCGA data as well as included detailed information in the tables as follows:

For TCGA dataset analysis, to identify TCGA breast tumors with EZH2 and cyclin E gene overexpression, we used data from normal breast tissue in TCGA and set up a cutoff at two standard deviation (SD) above median normal expression of EZH2. RNAseq dataset included a total 515 breast cancer samples and 102 normal breast tissues. The subtypes were defined by the *PAM50* gene signature. TCGA RNAseq data used this study were downloaded from Broad Institute Firehose website (<https://gdac.broadinstitute.org/>; BRCA cohort). A total of 515 breast cancer samples and with tumor subtype information and 102 normal breast tissues were used in our analysis (*Cancer Genome Atlas Network, Nature. 2012 Oct 4; 490: 61–70*). To call gene overexpression in breast tumors, we used an arbitrary cutoff threshold at two standard deviation above median normal expression of that gene (log₂ RSEM of 8.259, 5.736, and 6.925 for EZH2, CCNE1, and CCNE2, respectively). Tumors with expression level above a cutoff were considered as having an overexpression. We have added the description in the Tables 1–3.

Table 1. Overexpression of *EZH2* in breast cancers.

EZH2 OE	Basal-like (n = 91)	HER2 (n = 116)	Luminal A (n = 130)	Luminal B (n = 172)	Normal-like (n = 6)
Yes	86	95	81	60	1
No	5	21	49	112	5
% OE	94.5	81.9	62.3	34.9	16.7

TCGA RNAseq data were downloaded from Broad Institute Firehose website (<https://gdac.broadinstitute.org/>; BRCA cohort). An arbitrary cutoff threshold at two standard deviation (SD) above median normal expression of that gene (log₂ RSEM of 8.259 for *EZH2*). Tumor with expression level above 2 SD were considered as having overexpression (OE).

Point-by-Point Response to Reviewer's Comments

Table 2. Overexpression of *CCNE1* and *CCNE2* in breast cancers.

CCNE1 OE	Basal-like (n = 91)	HER2 (n = 116)	Luminal A (n = 130)	Luminal B (n = 172)	Normal-like (n = 6)
Yes	88	94	52	53	1
No	3	22	78	119	5
% OE	96.7	81.0	40.0	30.8	16.7

CCNE2 OE	Basal-like (n = 91)	HER2 (n = 116)	Luminal A (n = 130)	Luminal B (n = 172)	Normal-like (n = 6)
Yes	81	112	109	110	1
No	10	4	21	62	5
% OE	89.0	96.6	83.8	64.0	16.7

TCGA RNAseq Dataset were downloaded from Broad Institute Firehose website (<https://gdac.broadinstitute.org>; BRCA cohort). An arbitrary cutoff threshold at two SD above median normal expression of that gene (log₂ RSEM of 5.736 and 6.925 for *CCNE1* and *CCNE2*, respectively). Tumors with expression level above a cutoff were considered as having overexpression (OE).

Table 3. Co-overexpression of *EZH2* and cyclin E genes in breast cancers.

EZH2/CCNE1 OE	Basal-like (n = 91)	HER2 (n = 116)	Luminal A (n = 130)	Luminal B (n = 172)	Normal-like (n = 6)
Yes	84	82	43	33	1
% Co-OE	92.3	70.7	33.1	19.2	16.7

EZH2/CCNE2 OE	Basal-like (n = 91)	HER2 (n = 116)	Luminal A (n = 130)	Luminal B (n = 172)	Normal-like (n = 6)
Yes	78	93	75	54	0
% Co-OE	85.7	80.2	57.7	31.4	0

TCGA RNAseq Dataset were downloaded from Broad Institute Firehose website (<https://gdac.broadinstitute.org>; BRCA cohort). Co-overexpression of *EZH2* and *CCNE1* or *CCNE2* was analyzed as described in Tables 1 and 2.

Point #3: *ESR1* should be identified on the heat map for the RNA-seq data. In addition, the resolution of the heat map is too low to identify individual genes following zooming in on the image. The resolution should be corrected to allow identification of individual genes.

Response to Point #3: Thank you for your suggestion. The *ESR1* gene has been included in the heat map of the up- and down-regulated genes as shown in the following figure (Fig. 5b). In fact, we have quantitated the ER α mRNA level by qPCR and determined ER α protein in the cell lysates of the tumors by Western blotting prior to RNAseq (Fig. S, only shown in point-by-point response). Also, we apologize for the low-resolution heat map and have replaced the original figure with one that is clearer in the revised manuscript. The RNAseq data has been submitted to the GEO repository following the instruction of editorial policy. The data is available to the

Point-by-Point Response to Reviewer's Comments

public now (GSE132194; NCBI tracking system #20042815). We have also provided the GSE number in the revised manuscript.

Figure 5. Global gene expression alterations in TNBC regulated by the CDK2/EZH2 signaling axis. Total RNAs were isolated from the SUM-149 transplant tumor tissues treated with vehicle, EZH2i or CDK2i by using the RNeasy® Mini Kit (QIAGEN). RNAseq was performed by NOVOGENE (UC Davis) and *Homo sapiens* (GRCH37/hg19) was used as reference genome annotation. (b) Representative heat map of top-regulated genes by blockade of CDK2/EZH2 signal axis.

Point-by-Point Response to Reviewer's Comments

Figure S. Verification of tumor samples for RNAseq. (a) Total RNAs were isolated from the SUM-149 transplant tumor tissues treated with vehicle, EZH2i or CDK2i by using the RNeasy® Mini Kit (QIAGEN). ERα levels in the samples were determined by qPCR analysis. (b) The whole tumor tissue lysates were immunoblotted with anti-ERα and actin as a loading control.

Point #4: *Rather than the RNA-seq data being "made available upon request", it should be submitted to a public repository.*

Response to Point #4: As mentioned in our response to **Point #3**, we have submitted the RNAseq data to the GEO repository. The data is available to the public now (GSE132194; NCBI tracking system #20042815).

Reviewer #3 (Replacement Reviewer for Reviewer #2) Comments:

Transitioning between breast cancer subtypes during breast cancer development is an intriguing clinical observation. The understanding of the mechanisms that govern the TNBC phenotype has a clinical significance in guiding rationally designed treatment. In this manuscript by Hung et al. colleagues, the authors generated novel transgenic models (EZH2WT, EZH2T416D, and Tg-Neu;EZH2T416D) and demonstrated that hyperphosphorylation of T416 on EZH drives lineage fate of the basal-type breast cancer phenotype. The author further demonstrated that inhibition EZH2 or CDK2 using clinically available small molecule inhibitors reinitiated ERalpha expression in TNBCs and rendered TNBCs susceptible to anti-ER treatment. Overall, the data presented in this manuscript provide important mechanistic connection of EZH2-ERalpha axis in the vivo setting. Clinically, the implications of this finding could guide the future development of more effective combinatorial targeted therapies for TNBC patients.

This revised manuscript sufficiently answered the issues raised by the previous reviewer. However, there are additional minor comments need to be addressed or clarified before warranting a publication at Nat Communications.

Major comments:

Point-by-Point Response to Reviewer's Comments

1) *It is puzzling that T416A (phosphorylation-deficient) mutation showed no difference to EZH2 constructs on H3K27me3 (Fig. S2a). This argues against the dependency of 416 phosphorylation of EZH2's function in regulating ERalpha. This point should be discussed.*

2) *The Abcam ERalpha antibody (Cat#108398, as indicated in the author rebuttal letter) reacts with human ERalpha, not the mouse ERalpha, as indicated by Abcam. Although it is hard to exclude the possible cross-reaction without experimental evidence, the author should pay extra caution when using this antibody to claim mouse ERalpha expression. Moreover, the ERalpha antibody (Cat#108398) is a monoclonal antibody. It is confusing that the author stated in the main text line 238 that it is a polyclonal antibody and in the Figure legend of Fig. S3i. "... was immunoblotted with polyclonal anti-ERα antibody from Abcam."*

3) *The new Fig. S3d nicely demonstrated the T416 is a downstream target of CDK2i. It should be placed in the main Figure 2 instead of in the Supplementary Figures. Similarly, Fig. S4a and b are important data demonstrating the synergistic effects between two drugs. It needs to be in the main Figure 4.*

4) *In Table 4, it is not clear whether the vehicle group has always been used the reference (control) group when performing a chi-squared hypothesis testing. In addition, the author should indicate whether the p-value is generated from Fisher's exact test.*

5) *In the RNA-seq experiment, did authors observed an up-regulation of ESR1 (ERa) gene expression after EZH2i treatment?*

Minor comments:

1) *The color balance of CK14 and CK18 IHC staining (Fig. 1B, C) need to be improved. The high level of red background of the current picture makes very difficult to evaluate the results.*

2) *Since this is the first report of EZH2T416D transgenic mouse, more detailed description, e.g. genotyping primers and protocols, need to be provided.*

3) *Western blot validation of phosphor-CDK2 under Dinaciclib should be included in Fig. 2d for data consistency.*

4) *Fig 3e is not the true experimental result. It should be removed or be used as part of summary schematics.*

Authors' Response: We greatly appreciate the reviewer for the comments and great suggestions to improve the scientific merit of our manuscript. Below please find our responses to each of these comments and suggestions.

Major Point #1: *It is puzzling that T416A (phosphorylation-deficient) mutation showed no*

Point-by-Point Response to Reviewer's Comments

difference to EZH2 constructs on H3K27me3 (Fig. S2a). This argues against the dependency of 416 phosphorylation of EZH2's function in regulating ERalpha. This point should be discussed.

Response to Major Point #1: We thank the reviewer for pointing out this phenomenon. We have repeated the experiments several times and have consistently observed similar results. T47D is a luminal type of breast cancer cells, and CDK2/cyclin E is lower in non-TNBCs than in TNBC tumors⁷ (Table 3). These results suggested that in luminal type breast cancer cells, the activity of CDK2 is lower, and therefore, phosphorylation level of EZH2^{WT} at T416 is also low, leading to less abundance of H3K27Me3. Under such circumstance, mutation of Thr 416 to Ala may not significantly change global H3K27me3 level. In contrast, the T416D mutation mimicked T416 phosphorylation (CDK2 independent), and the EZH2^{T416D} mutant exhibited the highest EZH2 activity and H3K27me3 level. This phenomenon actually reflects the finding in Figure S2b in which both EZH2^{WT} and EZH2^{T416A} did not lead to tumor development in xenograft mouse model, which contrasts with the T416D mutant. We have added this discussion in the revised manuscript (pages 7–8).

Figure S2. Ectopic expression of EZH2^{T416D} but not EZH2^{T416A} in T47D cells switches luminal cancer cells into basal-like type and enhances tumorigenesis independent of estrogen. (a) Ectopic expression of EZH2 T416 phospho-mimick T416D in luminal breast cancer cells downregulates ERα and upregulates H3K37Me3. (b) Expression of EZH2T416D in non-tumorigenic breast cancer cells in NOD/SCID mice led to tumor development. WT, EZH2^{WT}; T416A, EZH2^{T416A}; T416D, EZH2^{T416D}.

Major Point #2: *The Abcam ERalpha antibody (Cat#108398, as indicated in the author rebuttal letter) reacts with human ERalpha, not the mouse ERalpha, as indicated by Abcam. Although it is hard to exclude the possible cross-reaction without experimental evidence, the author should pay extra caution when using this antibody to claim mouse ERalpha expression. Moreover, the ERalpha antibody (Cat#108398) is a monoclonal antibody. It is confusing that the author stated in the main text line 238 that it is a polyclonal antibody and in the Figure legend of Fig. S3i. "... was immunoblotted with polyclonal anti-ERα antibody from Abcam."*

Response to Major Point #2: We deeply appreciate the reviewer's comment on the antibody against ERα we used in the experiments. We apologize for the typo in which we indicated that the antibody is polyclonal. We have corrected this (page 9 of revised manuscript) to indicate that this is a **rabbit monoclonal antibody**. We compared many commercial products of anti-

Point-by-Point Response to Reviewer's Comments

ER α antibody and identified one from Abcam (Cat# 108398) as the best one as it recognizes both human and mouse ER α (see datasheet from Abcam). This antibody can be only used for Western blotting for cells of mouse origin.

[REDACTED]

Major Point #3: *The new Fig. S3d nicely demonstrated the T416 is a downstream target of CDK2i. It should be placed in the main Figure 2 instead of in the Supplementary Figures. Similarly, Fig. S4a and b are important data demonstrating the synergistic effects between two drugs. It needs to be in the main Figure 4.*

Response to Major Point #3: We thank the reviewer's suggestions. In the revised manuscript we moved Figure S3d to main Figure 2 h and Figure S4a and c to main Figure 4 b and c.

Major Point #4: *In Table 4, it is not clear whether the vehicle group has always been used the reference (control) group when performing a chi-squared hypothesis testing. In addition, the author should indicate whether the p-value is generated from Fisher's exact test.*

Point-by-Point Response to Reviewer's Comments

Response to Major Point #4: We appreciate the reviewer for pointing out. The vehicle group was consistently used as the control group. Comparison of proportions between the groups were analyzed by using MedCalc software ("N-1" Chi-squared test). The p-values in Table 4 were calculated using comparison of proportions calculator (**MedCalc**) and not generated from Fisher's exact test. The statistical analysis and references have been updated in the Methods section and Table 4 in the revised manuscript (page 24).

Table 4. Inhibition of CDK2/EZH2 signaling axis reduces tumor lung metastasis.

Group	# Lung Metastasis	# Total Mice	Metastatic rate (%)	p value*
Vehicle	8	8	100	–
TAM	8	8	100	NS
EPZ	4	8	50.0	0.0253
EPZ+TAM	3	8	37.5	0.0090
DINA	4	7	57.1	0.0454
DINA+TAM	4	8	50	0.0253

*Statistical analysis was performed by using MedCalc software ("N-1" Chi-squared test)^{55,56}. A p value < 0.05 is defined as statistically significant. EPZ, EPZ-6438; DINA, Dinaciclib; TAM, tamoxifen. NS, Not significant.

Major Point #5: *In the RNA-seq experiment, did authors observed an up-regulation of ESR1 (ER α) gene expression after EZH2i treatment?*

Response to Major Point #5: Indeed, *ESR1* expression was increased after both CDKi and EZH2i treatment. The *ESR1* gene was included in the heat map of the up- and down-regulated genes as shown in the following figure (Fig. 5b). As we mentioned above, we did quantitate the ER α mRNA level by qPCR and determined ER α protein levels in the cell lysates of the tumors by Western blotting prior to RNAseq (Fig. S, only shown in point-by-point response). We have replaced the original Figure 5b a higher-resolution version in the revised manuscript. The RNAseq data was submitted to the GEO repository following the editorial policy. The data is available to the public now (GSE132194; NCBI tracking system #20042815).

Figure 5. Global gene expression alterations in TNBC regulated by the CDK2/EZH2 signaling axis. Total RNAs were isolated from the SUM-149 transplant tumor tissues treated with vehicle, EZH2i or CDK2i by using the RNeasy® Mini Kit (QIAGEN). RNAseq was performed by NOVOGENE (UC Davis) and *Homo sapiens* (GRCH37/hg19) was used as reference genome annotation. (b) Representative heat map of top-regulated genes by blockade of CDK2/EZH2 signal axis.

Point-by-Point Response to Reviewer's Comments

Figure S. Verification of tumor samples for RNAseq. (a) Total RNAs were isolated from the SUM-149 transplant tumor tissues treated with vehicle, EZH2i or CDK2i by using the RNeasy® Mini Kit (QIAGEN). ER α levels in the samples were determined by qPCR analysis. (b) Whole tumor cell lysates were immunoblotted with anti-ER α and actin as a loading control.

Minor Point #1: *The color balance of CK14 and CK18 IHC staining (Fig. 1B, C) need to be improved. The high level of red background of the current picture makes very difficult to evaluate the results.*

Response to Minor Point #1: Based on the reviewer's comments, we have replaced the photos with newer ones with improved color balance in the revised manuscript.

Figure 1. Targeted expression of EZH2^{T416D} mutant in mammary epithelia leads to mammary gland tumor with basal-like lineage type. (b and c) Representative images of IHC staining of the tumor sections from the primary tumors of different genotypes of GEMMs. Antibodies against CK18 and CK14, the luminal and basal-like lineage marker, were used for staining, the tumor slides from Tg-Neu, p53^{f/f};Brca1^{f/f};LGB-Cre and Tg-C3 mice were used for luminal and basal-like positive control, respectively.

Minor Point #2: *Since this is the first report of EZH2^{T416D} transgenic mouse, more detailed description, e.g. genotyping primers and protocols, need to be provided.*

Response to Minor Point #2: We thank the reviewer for the suggestions and have added detailed description in Methods section as shown below:

Point-by-Point Response to Reviewer's Comments

Generation of EZH2 mutant transgenic mouse model. MMTV-EZH2T416D transgenic mouse was generated by cloning a cDNA harboring T416D mutation into the EcoRI/HindIII of plasmid MMTV-SV40-BSSK. The cDNA of human EZH2 was sequenced prior to digestion with SpeI and Sall. The purified DNA fragments containing the expression cassette of EZH2 mutant was injected into one-cell zygotes of FVB/N mice performed in the Genetically Engineered Mouse Core facility, MD Anderson Cancer Center. MMTV-EZH2WT mice were generated as previous described¹². The transgenic founder mice were confirmed by PCR-based genotyping and Western blot analysis of mammary epithelial cells with EZH2 and H3K27Me3 antibodies. Genotyping of the EZH2T416D transgene was performed by PCR with two pairs of primers:

Pair 1 Forward: 5'-GGA ACC TTA CTT CTG TGG TGT-3'
Reverse: 5'-GCA TTC CAC CAC TGC TCC CAT TC-3'

Pair 2 Forward: 5'-CTG ATC TGA GCT CTG AGT-3'
Reverse: 5'-GAC AAA CAA TTC CAA CAG TC-3'

Minor Point #3: *Western blot validation of phospho-CDK2 under Dinaciclib should be included in Fig. 2d for data consistency.*

Response to Minor Point #3: We thank the reviewer for the comment and performed the suggested experiment. Our results indicated that along with reduced p-T160 CDK2 level in TNBC cells, CDK2i treatment indeed increased ER α in a dose-dependent manner (Fig. 2e). The results suggested that reduction of CDK2 activity leads to decreased p-EZH2 and reactivated ER α gene expression in TNBC cells.

Figure 2d. Inhibition of CDK2 activity in TNBCs upregulates ER α protein. MDA-MB231 cells in Matrigel 3D culture were treated with dinaciclib (DINA) at different concentrations for 3 days. Whole cell lysates of the treated cells were immunoblotted with specific antibodies as indicated. (d) pT416-EZH2 (p-EZH2) was probed using specific monoclonal antibody, and (e) polyclonal antibody against pT160-CDK2 was used to detect activated CDK2, pCDK2.

Response to Minor Point #4: *Fig 3e is not the true experimental result. It should be removed or be used as part of summary schematics.*

Point-by-Point Response to Reviewer's Comments

Author response: We appreciate your suggestion and have moved this diagram to Supplementary Figure S6c to as a working model to summarize our work.

Fig.S6c. Working model. (Left) CDK2-mediated specific site T416 phosphorylation is required for guiding the PRC2 complex to bind to the lineage-specific promoters such as ER α and GATA3.

pT416-EZH2 containing PRC2 complex biasedly silences ER α and GATA3 expression; (Right) Blockade of CDK2/EZH2 signaling axis by inhibitors reactivates ER α and luminal gene expressions.

REVIEWERS' COMMENTS:

Reviewer #1 (Remarks to the Author):

All concerns for this reviewer have been appropriately addressed.

Reviewer #3 (Remarks to the Author):

The authors have sufficiently addressed the reviewer's comments. The revised manuscript has been significantly improved.